# Comprehensive Review of Vision-Based Fall Detection Systems

**DOI:** 10.3390/s21030947

**Published:** 2021-02-01

**Authors:** Jesús Gutiérrez, Víctor Rodríguez, Sergio Martin

**Affiliations:** 1Universidad Nacional de Educación a Distancia, Juan Rosal 12, 28040 Madrid, Spain; smartin@ieec.uned.es; 2EduQTech, E.U. Politécnica, Maria Lluna 3, 50018 Zaragoza, Spain; victorhugo@invi.uned.es

**Keywords:** artificial vision, neural networks, fall detection, fall characterization, fall classification, fall dataset

## Abstract

Vision-based fall detection systems have experienced fast development over the last years. To determine the course of its evolution and help new researchers, the main audience of this paper, a comprehensive revision of all published articles in the main scientific databases regarding this area during the last five years has been made. After a selection process, detailed in the Materials and Methods Section, eighty-one systems were thoroughly reviewed. Their characterization and classification techniques were analyzed and categorized. Their performance data were also studied, and comparisons were made to determine which classifying methods best work in this field. The evolution of artificial vision technology, very positively influenced by the incorporation of artificial neural networks, has allowed fall characterization to become more resistant to noise resultant from illumination phenomena or occlusion. The classification has also taken advantage of these networks, and the field starts using robots to make these systems mobile. However, datasets used to train them lack real-world data, raising doubts about their performances facing real elderly falls. In addition, there is no evidence of strong connections between the elderly and the communities of researchers.

## 1. Introduction

In accordance with the UN report on the aging population [1], the global population aged over 60 doubled its number in 2017 compared to 1980. It is expected to double again by 2050 when they exceed the 2 billion mark. By this time, their number will be greater than the number of teenagers and youngsters aged 10 to 24.

The phenomenon of population aging is a global one, more advanced in the developed countries, but also present in the developing ones, where two-thirds of the worlds older people live, a number which is rising fast.

With this perspective, the amount of resources devoted to elderly health care is increasingly high and could, in the non-distant future, become one of the most relevant world economic sectors. Because of this, all elderly health-related areas have attracted great research attention over the last decades.

One of the areas immersed in this body of research has been human fall detection, as, for this community, over 30% of falls cause important injuries, ranging from hip fracture to brain concussion, and a good number of them end up causing death [2].

The number of technologies used to detect falls is wide, and a huge number of systems able to work with them have been developed by researchers. These systems, in broad terms, can be classified as wearable, ambient and camera-based ones [3].

The first block, the wearable systems, incorporate sensors carried by the surveilled individual. The technologies used by this group of systems are numerous, ranging from accelerometers to pressure sensors, including inclinometers, gyroscopes or microphones, among other sensors. R. Rucco et al. [4] thoroughly review these systems and study them in-depth. In this article, systems are classified in accordance with the number and type of sensors, their placement and the characteristics of the study made during the system evaluation phase concluding that most systems incorporate one or two accelerometric sensors attached to the trunk.

The second block includes systems whose sensors are placed around the monitored person and include pressure, acoustic, infra-red, and radio-frequency sensors. The last block, the object of this review, groups systems able to identify falls through artificial vision.

In parallel, over the last years, artificial vision has experienced fast development, mainly due to the use of artificial neural networks and their ability to recognize objects and actions.

This artificial vision development applied to human activity recognition in general, and human fall detection in particular, has given very fruitful outcomes in the last decade.

However, up to where we know, no systematic reviews on the specific area of vision-based detection systems have been made, as all references to this field have been included in generic fall detection system reviews.

This review intends to shed some light on the process of development followed by vision-based fall detection systems, so researchers get a clear image of what has been done in this field during the last five years that help them in their investigation process. In this study, authors intend to show the main advantages and disadvantages of all processes and algorithms used in the reviewed systems so new developers get a clear picture of the state of the art in the field of human fall detection through artificial vision, an area that could significantly improve living standards for the dependent community and have a high impact on their day-to-day lives.

The article is organized as follows: In Section 2, Materials and Methods, characterization and classification techniques are described and applied to the preselected systems, so a number of them are finally declared as eligible to be included in this review. In Section 3, Results, those systems are presented and roughly described, the databases used for their validation are presented, and some performance comparisons are made. In the next Section, Discussion, the algorithms and processes used by the systems are described and, in the last part of the review, Section 5, conclusions are extracted based on all the previously presented information.

## 2. Materials and Methods

In this paper, we focus on artificial vision systems able to detect human falls. To fulfill this purpose, we have performed a deep review of all published papers present in public databases of research documentation (ScienceDirect, IEEE Explorer, Sensors database). This documental search was based on different text string searches and was executed from May 2020 to December 2020. The time frame of publication was established between 2015 and 2020, so the last developments in the field can be identified, and the study serves to orientate new researchers. The terms used in the bibliographical Boolean exploration were “fall detection” and “vision”. A secondary search was carried out to complete the first one by using other search engines of scholarly literature focused on health (PubMed, MedLine). All searches have been limited to articles and publications in English, language used by most area researchers.

After an initial analysis of papers fulfilling these searching criteria 81 articles, describing the same number of systems were selected. They illustrate how fall detection systems based on artificial vision have evolved in the last five years.

The selection process included an initial screening made through reference management software to guarantee no duplication, and a manual screening, whose objective was making sure the article covered the field, did not fall within the field of the fall prevention or human activity recognition (HAR), did not mix vision technologies with other ones and were not studies intending to classify the human gait as an indicator of fall probability. This way, the review is purely centered on artificial vision fall detection.

The entire process is summarized in the flow diagram shown in Figure 1.

All selected systems were studied one-by-one to determine their characterization and classification techniques, describing them in-depth in the Discussion (Section 4), so a full taxonomy can be made based on their characteristics. In addition, performance comparisons are also included, so conclusions on which ones are the most suitable systems can be reached.

## 3. Results

The article search and selection process started with an initial identification of 929 potential articles. Duplicated ones and those whose title clearly did not match the required content were discarded, leaving 430 articles that were assessed for eligibility. These articles were then reviewed, and those related to HAR, fall prevention, mixed technologies, gait studies and the ones which did not cover the area of vision-based fall detection were discarded, so; finally, 81 articles are considered in the review.

The selected systems were thoroughly revised and classified in accordance with the used characterization and classification methods, as well as the employed type of signal. The used dataset for performance determination and its indicators values have also been studied. All this information is included in Table 1.

System comparison data were used to develop Table 2, and finally, all main characteristics of publicly accessible datasets used by any of the systems are included in Table 3.

## 4. Discussion

The studied systems illustrate visual-based fall detection evolution in the last five years. These systems follow a parallel path to other human activity recognition systems, with increasingly intense use of artificial neural networks (ANN) and a clear tendency towards cloud computing systems, except for the ones mounted on robots.

All studied systems follow, with nuances, a three-step approach to fall detection through artificial vision.

The first step, introduced in Section 4.1 and not always needed, includes video signal preprocessing in order to optimize it as much as possible.

Characterization is the second step, studied in Section 4.2, where image features are abstracted, so what happens in the images can be expressed in the form of descriptors that will be classified in the last step of the process.

The third process step, explained in Section 4.3, intends to tag the observed actions, which main features are characterized by abstract descriptors, as a fall event or one which is not, so measures can be taken to help the fallen person as fast as possible.

Some of the studied systems follow a frame-by-frame approach where the sole system goal is classifying human pose as fallen or not, leaving aside the fall motion itself. For those systems trying to determine if a specific movement may be a fall, silhouette tracking is a basic support operation developed through different processes. Tracking techniques used by the studied systems are explained in Section 4.4.

Finally, a comparison in classifying algorithm performance and validation datasets is presented in Section 4.5 and Section 4.6.

### 4.1. Preprocessing

The final objective of this phase is either distortion and noise reduction or format adaptation, so downstream system blocks can extract characteristic features with classification purposes. Image complexity reduction could also be an objective during the preprocessing phase in some systems, so the computational cost can be reduced, or video streaming bandwidth use can be diminished.

The techniques grouped in this Section for decreasing noise are numerous and range from Gaussian smoothing used in [31] to the morphological operations executed in [17,31,74] or [24]. They are introduced in subsequent Section as a part of the foreground segmentation process.

Format adaptation processes are present in several of the studied systems, as is the case in [48], where images are converted to grayscale and have their histograms equalized before being transferred to the characterization process.

Image binarization, as in [89], is also introduced as a part of the systematic effort to reduce noise during the segmentation process, while some other systems, like the one presented in [56], pursue image complexity decreasing by transforming video signals from red, green and blue (RGB) to black and white and then applying a median filter, an algorithm which assigns new values to image pixels based on the median of the surrounding ones.

Image complexity reduction is a goal pursued by some systems, as the one proposed in [91], which introduces compressed sensing (CS), an algorithm first proposed by Donoho et al. [107] used in signal processing to acquire and reconstruct a signal. Through this technique, signals, sparse in some domain, are sampled at rates much lower than required by the Nyquist–Shannon sampling theorem. The system uses a three-layered approach to CS by applying it to video signals, which allows privacy preservation and bandwidth use reduction. This technique, however, introduces noise and over-smooths edges, especially those in low contrast regions, leading to information loss and image low-resolution. Therefore, image complexity reduction feature characterization often becomes a challenge.

### 4.2. Characterization

The second process step intends to express human pose and/or human motion as abstract features in a qualitative approach, to quantify their intensity in an ulterior quantity approach. These quantified features are then used with classifying purposes in the last step of the fall detection system.

These abstract pose/action descriptors can globally be classified into three main groups: global, local and depth.

Global descriptors analyze images as a block, segmenting foreground from background, extracting descriptors that define it and encoding them as a whole.

Local descriptors approach the abstraction problem from a different perspective and, instead of segmenting the block of interest, process the images as a collection of local descriptors.

Depth characterization is an alternative way to define descriptors from images containing depth information by either using depth maps or skeleton data extracted from a joint tracking process.

#### 4.2.1. Global

Global descriptors try to extract abstract information from the foreground once it has been segmented from the background and encode it as a whole.

This kind of activity descriptors was very commonly used in artificial vision approaches to human activity recognition in general and to fall detection in particular. However, over time, they have been displaced by local descriptors or used in combination with them, as these ones are less sensitive to noise, occlusions and viewpoint changes.

Foreground segmentation is executed in a number of different ways. Some approaches to this concept establish a specific background and subtract it from the original image; some others locate regions of interest by identifying the silhouette edges or use the optical flow, generated as a consequence of body movements, as a descriptor. Some global characterization methods segment the human silhouette over time to form a space–time volume which characterizes the movement. Some other methods extract features from images in a direct way, as in the case of the system described in [48], where every three frames, the mean square error (MSE) is determined and used as an indicator of image similarity.

##### Silhouette Segmentation

Human shape segmentation can be executed through a number of techniques, but all of them require background identification and subtraction. This process, known as background extraction, is probably the most visually intuitive one, as its product is a human silhouette.

Background estimation is the most important step of the process, and it is addressed in different ways.

In [17,24,56,74], as the background is supposed constant, an image of it is taken during system initialization, and a direct comparison allows segmentation of any new object present in the video. This technique is easy and powerful; however, it is extremely sensitive to light changes. To mitigate this flaw, the system described in [31], where the background is also supposed stable, a median throughout time is calculated for every pixel position in every color channel. Then, it is directly subtracted from the observed image frame-by-frame.

Despite everything, the obtained product still contains a substantial amount of noise associated with shadows and illumination. To reduce it, morphological operators can be used as in [17,24,31,74]. Dilation and/or erosion operations are performed by probing the image at all possible places with a structuring element. In the dilation operation, this element works as a local maximum filter and, therefore, adds a layer of pixels to both inner and outer boundary areas. In erosion operations, the element works as a local minimum filter and, as a consequence, strips away a layer of pixels from both regions. Noise reduction after segmentation can also be performed through Kalman filtering, as in [92], where this filtering method is successfully used with this purpose.

An alternative option for background estimation and subtraction is the application of Gaussian mixture models (GMM), a technique used in [7,11,14,78,92], among others, that models the values associated with specific pixels as a mix of Gaussian distributions.

A different approach is used in [6], where the Horprasert method [108] is applied for background subtraction. It uses a computational color model that separates the brightness from the chromaticity component. By doing it, it is possible to segment the foreground much more efficiently when light disturbances are present than with previous methods, diminishing this way light change sensitiveness. In this particular system, pixels are also clustered by similarity, so computational complexity can be reduced.

Some systems, like the one presented in [7], apply a filter to determine silhouette contours. In this particular case, a Sobel filter is used, which determines a two-dimensional gradient of every image pixel.

Other segmentation methods, like vibe [19], used in [22,94], store, associated with specific pixels, previous values of the pixel itself and its vicinity to determine whether its current value should be categorized as foreground or background. Then, the background model is adapted by randomly choosing which values should be substituted and which not, a clearly different perspective from other techniques, which give preference to new values. On top of that, pixel values declared as background are propagated into neighboring pixels part of the background model.

The system in [8] segments the foreground using the technique proposed in [109], where the optical flow (OF), which are presented in later Sections, is calculated to determine what objects are in motion in the image, feature used for foreground segmentation. In a subsequent step, to reduce noise, images are binarized and morphological operators are applied. Finally, the points marking the center of the head and the feet are linked by lines composing a triangle whose area/height ratio will be used as the characteristic classification feature.

Some algorithms, like the illumination change-resistant independent component analysis (ICA), proposed in [95], combine features of different segmentation techniques, like GMM and self-organizing maps, a well-known group of ANN able to classify into low dimensional classes very high dimensional vectors, to overcome the problems of silhouette segmentation associated with illumination phenomena. This algorithm is able to successfully tackle segmentation errors associated with sudden illumination changes due to any kind of light source, both in images taken with omnidirectional dioptric cameras and in plain ones.

ICA and vibe are compared in [94] by using a dataset specifically developed for that system with better results for the ICA algorithm.

In [9], foreground extraction is executed in accordance with the procedure described in [110]. This method integrates the region-based information on color and brightness in a codeword, and the collection of all codewords are grouped in an entity called codebook. Pixels are then checked in every single new frame and, when its color or brightness does not match the region codeword, which encodes area brightness and color bands, it is declared as foreground. Otherwise, the codeword is updated, and the pixel is declared as area background. Once pixels are tagged as foreground, they are clustered together, and codebooks are updated for each one of them. Finally, these regions are approximated by polygons.

Some systems, like the one in [9], use orthogonal cameras and fuse foreground maps by using homography. This way, noise associated with illumination variations and occlusion is greatly reduced. The system also calculates the observed polygon area/ground projected polygon area rate as the main feature to determine whether a fall event has taken place.

Self-organizing maps is a technique, well described in [111], used with segmentation purposes in [58]. When applied, initial background estimation is made based on the first frame at system startup. Every pixel of this initial image is associated with a neuron in an ANN through a weight. Those weights are constantly updated as new frames flow into the system and, therefore, the background model changes. Self-organizing maps have been successfully used to subtract foreground from background, and they have proved a good resilience to the light variation noise.

Binarization is a technique used for background subtraction, especially in infrared (IR) systems, as the one presented in [89], where the inputs IR signals pixels are assigned two potential values, 0 and 1. All pixels above a certain threshold value are assigned a value 1 (human body temperature dependent), and all others are given a value of 0. This way, images are expressed in binary format. However, the resulting image usually has a great amount of noise. To reduce it, the algorithm is able to detect contours through gradient determination. Pixels within closed contours whose dimensions are close to the ones of a person continue being assigned a value 1, while the rest are given a value 0.

Once the foreground has been segmented, it is time to characterize it through abstract descriptors that can be classified at a later step.

This way, after background subtraction, features used for characterization in [31] and [14] are silhouettes eccentricity, orientation and acceleration of the ellipse surrounding the human shape.

Characteristic dimensions of the bounding box surrounding the silhouette are also a common distinctive feature, as is the case in [78]. In [67], a silhouette’s horizontal width is estimated at 10 vertically equally spaced points, and, in [74], five regions are defined in the bounding box, being its degree of occupancy by the silhouette is used as the classifying element.

Other features also used for characterization used in [7,39] include Hu moments, a group of six image moments in variables to translation, scale, rotation, and reflection, plus a seventh one, which changes sign for image reflection. These moments, assigned to a silhouette, do not change as a result of the point of view alterations associated with body displacements. However, they dramatically vary as a result of human body pose changes as the ones associated with a fall. This way, a certain resistance to noise due to the point of view change is obtained.

The Feret diameter, the distance from the two most distant points of a closed line when taking a specific reference orientation, is another used distinctive feature. The system described in [58] uses this distance, with a reference orientation of 90°, to characterize the segmented foreground.

Procrustes analysis is a statistical method that uses minimum square methods to determine the needed similarity transformations required to adjust two models. This way, they can be compared, and a Procrustes distance, which quantifies how similar the models are, can be inferred. This distance, employed in some of the studied systems as a characterization feature, is used to determine similarities between silhouettes in consecutive frames and, therefore, as a measure of its deformation as a result of pose variation.

The system introduced in [22], after identifying in each frame the torso section in the segmented silhouette, stores its position in the last 100 frames in a database and uses this trajectory as a feature for fall recognition.

To decrease sensitiveness to noise as a result of illumination noise and viewpoint changes, some systems combine RGB global descriptors and depth information.

This is the case of [49], where the system primarily uses depth information, but when it is not available, RGB information is used instead. In that case, images are converted to grayscale and pictures are formed by adding up the difference between consecutive frames. Then, features are extracted at three levels. At the pixel level, where gradients are calculated, at the patch level, where adaptive patches are determined, and at the global level, where a pyramid structure is used to combine patch features from the previous level. The technique is fully described in [112].

A different approach to the same idea is tried in [63], where depth information is derived from monocular images as presented in [12]. This algorithm uses monocular visual cues, such as texture variations, texture gradients, defocus and color/haze. It mixes all these features with range information derived from a laser range finder to generate, through a Markov random field (MRF) model, a depth map. This map is assembled by splitting the image into patches of similar visual cues and assigning them depth information that is related to the one associated with other image patches. Then, and to segment foreground from background, as the human silhouette has an almost constant depth, a particle swarm optimization (PSO) method is used to discover the optical window in which the variance of the image depth is minimum. This way, patches whose depth information is within the band previously defined are segmented as foreground.

This method, first introduced in [113], was designed to simulate collective behaviors like the ones observed in flocks of birds or swarms of insects. It is an iterative method where particles progressively seek optimal values. This way, in every iteration, depth values with the minimum variance associated with connected patches are approximated, increasing until an optimal value is reached.

##### Space–Time Methods

All previously presented descriptors abstract information linked to specific frames and, therefore, they should be considered as static data, which clustered along time, acquire a dynamic dimension.

Some methods, however, present visual information where the time component is already inserted and, therefore, dynamic descriptors could be inferred from them.

That is the case of the motion history image (MHI) process. Through this method, after silhouette segmentation, a 2-D representation of its movement, which can be used to estimate if the movement has been fast or slow, is built up. It was first introduced by Bobick et al. [114] and reflects motion information as a function of pixel brightness. This way, all pixels represent moving objects bright with an intensity function of how recent movement is. This technique is used in [16,17,92] to complement other static descriptors and introduce the time component.

Some systems, like the one introduced in [41], split the global MHI feature in sub-MHIs that are linked to the bounding boxes created to track people. This way, a global feature like MHI is actually divided into parts, and the information contained in each one of them is associated with the specific silhouette responsible for the movement. Through this procedure, the system is able to locally capture movement information and, therefore, able to handle several persons at the same time.

##### Optical Flow

Optical flow (OF) can be defined as the perceived motion of elements between two consecutive frames of a video clip resulting from the relative changes in angle and distance between the objects and the recording camera.

OF, as MHI, is a characterization feature that integrates the time dimension in the information abstraction process and, therefore, a dynamic descriptor.

A number of methods to obtain OF have been developed, being the Lucas–Kanade–Tomasi (LKT) feature tracker, presented in [115,116], the most used one. This is the OF obtaining procedure used in all the studied systems which use this feature as a dynamic descriptor.

Two main approaches are considered to obtain OF, sparse, where only relevant points are followed, and dense, where all image pixels are taken into consideration to collect their flow vectors.

In [17,24,32,75,83,86,88], a dense OF is created that will be used as one of the image characteristic features from which descriptors can be extracted.

Some of these systems obtain OF from segmented objects, as is the case in [17], where, after silhouette segmentation, an OF is derived, and its motion co-occurrence feature (MCF), which is the modulus/direction histogram of the OF, is used for classification.

The system in [24] also extracts a dense OF from segmented objects. In this case, after OF determination, it distributes flow vectors on a circle in accordance with their direction. The resulting product is a Von Mises distribution of the OF flow vectors, which is used as the characterization feature for classification.

In some of the studied systems, like the one presented in [83], the dense optical flow is used as the input of a neural network to generate movement descriptors.

In [22], a sparse OF of relevant points on the silhouette edge is derived, and their vertical velocity will be used as a relevant descriptor for fall identification.

OF has proven to be a very robust and effective procedure to segment the foreground, especially in situations where backgrounds are dynamic, as is the case in fall detection systems mounted on robots that patrol an area searching for fallen people.

##### Feature Descriptors

Local binary patterns (LBP), as used in [18], is an algorithm for feature description. In this technique, an operator iterates over all image pixels and thresholds its neighborhood with the pixel’s own value. This way, a binary pattern is composed. Occurrence histograms based on resulted binary patterns of the entire image, or a part of it, are used as feature descriptors.

Local binary pattern histograms from three orthogonal planes (LBP-TOP) are a further development of the LBP concept. They incorporate time and, therefore, movement in the descriptor, transforming it into a dynamic one. This technique computes each pixel LBP over time, building, this way, a three-dimensional characterization of the video signal by integrating space and temporal properties.

The system described in [91] takes, as input for characterization, a video signal which has gone through multilayered compressed sensing (CS) algorithm and that, therefore, has lost information, especially in low contrast areas. To overcome this difficulty, the system obtains the optical flow of the video signal after the CS process has taken place, and the LBP-TOP is applied over that OF, highly improving the characterization this way. As the video quality is so poor, the OF extraction based on pixel motion is ineffective. To obtain it, low-rank and sparse decomposition theory, also known as robust principal component analysis (RPCA) [117], is used to reduce noise. This technique is a modification of the statistical method of principal component analysis whose main objective is to separate, in a corrupted signal, a video one, in this case, the real underlying information contained in the original image from the sparse errors introduced by the CS process.

The histogram of oriented gradients (HOG), as used in [18], is another feature descriptor technique introduced by N. Dalal et al. [118] in the field of human detection with success. The algorithm works over grayscale images using edge detection to determine object positions. This approach uses gradient as the main identification feature to establish where body edges are. It takes advantage of the fact that gradients will sharply rise at body edges in comparison with the mean gradient variation of the area they are placed in. To identify those boundaries, a mask is applied on each pixel and gradients are determined through element-wise multiplication. Histograms of gradient orientation are then created for each block, and, in the final stages of the process, they are normalized both locally and globally. These histograms are used as image feature descriptors.

The system proposed in [71] incorporates HOGs as the image descriptor, which, in later stages of the identification algorithm, is used by an ANN to determine whether a fall has occurred.

#### 4.2.2. Local

Local descriptors approach the problem of pose and movement abstraction in a different way. Instead of segmenting the foreground and extracting characteristic features from it, encoding them as a block, they focus on area patches from which relevant local features, characteristic of human movement or human pose, can be derived.

Over time, local descriptors have substituted or complemented global ones, as they have proofed to be much more resistant to noise or partial occlusion.

Characterization feature techniques focused on fall detection, pay attention to head motion, body shape changes and absence of motion [119]. The system introduced in [81] uses the two first groups of features. It models body shape changes and head motion by using the extended CORE9 framework [120]. This framework uses minimum bounding rectangles to abstract body movements. The system slaves bounding boxes to legs, hands and head, which is taken as the reference element. Then, directional, topological, and distance relations are established between the reference element and the other ones. All this information is finally used for classification purposes.

The vast majority of studied systems that implement local descriptors do it through the use of ANNs. ANNs are a major research area at the moment, and their application to the artificial vision and human activity recognition is a hot topic. These networks, which simulate biological neural networks, were first introduced by Rosenblatt [121] through the definition of the perceptron in 1958.

There are two main families of ANNs with application in artificial vision, human pose estimation and human fall detection, which have been identified in this research. These two families are convolutional neural networks (CNN) and recurrent neural networks (RNN).

ANNs are able to extract feature maps out of input images. These maps are local descriptors able to characterize the different local patches that integrate an image.

RNNs are connectionist architectures able to grasp the dynamics of a sequence due to cycles in its structure. Introduced by Hopfield [122], they retain information from previous states and, therefore, they are especially suitable to work with sequential data when its flow is relevant. This effect of information retention through time is obtained by implementing recurrent connections that transfer information from previous time steps to either other nodes or to the originating node itself.

Among RNNs architectures, long short-term memory (LSTM) ones are especially useful in the field of fall detection. Introduced by Hochreiter [123], LSTMs most characteristic feature is the implementation of a hidden layer composed of an aggregation of nodes, called memory cells. These items contain nodes with a self-linked recurrent connection, which guarantees information will be passed along time with no vanishing. Unlike other RNNs, whose long-term memory materializes through weights given to inputs, which change slowly during training, and whose short-term memory is implemented through ephemeral activations, passed from a node to the successive one, LSTMs introduce an intermediate memory step in the memory cells. These elements internally retain information through their self-linked recurrent connections, which include a forget gate. Forget gates allow the ANN to learn how to forget the contents of previous time steps.

LSTM topologies, like the one implemented in [77], allow the system to recall distinctive features from previous frames, incorporating, this way, the time component to the image descriptors. In this particular case, an RNN is built by placing two LSTM layers between batch normalization layers, whose purpose is to make the ANN faster. Finally, a last layer of the network, responsible for classification, implements a Softmax algorithm.

Some LSTMs architectures, like the one described in [71], are used to determine characteristic foreground features. This ANN is able to establish a silhouette center and establish angular speed, which will be used as a reference to determine whether a fall event has taken place.

The system proposed in [76] includes several LSTM layers. This encoding-decoding architecture integrates an encoding block, which encodes the input data, coming from a CNN block used to identify joints and estimate body pose, to a vector of fixed dimensionality, and a decoding block, composed of a layer able to output predictions on future body poses. This architecture is based on the seq2seq model proposed in [124] and has been successfully used in this system with prediction purposes, substantially reducing fall detection time as it is assessment is made on a prediction, not on observation.

A specific LSTM design is the bidirectional one (Bi-LSTM). This architecture integrates two layers of hidden nodes connected to inputs and outputs. Both layers implement the idea of information retention through time in a different way. While the first layer has recurrent connections, in the second one, connections are flipped and passed backward through the activation function signal. This topology is incorporated in [104], where Bi-LSTM layers are stacked over CNN layers used to segment incoming images.

CNNs were inspired by the neural structure of the mammal visual system, very especially by the patterns proposed by Hubel et al. [125]. The first neural network model with visual pattern recognition capability was proposed by Fukushima [126], and, based on it, LeCun and some collaborators developed CNNs with excellent results in pattern recognition, as shown in [127,128].

This family of ANNs is assembled by integrating three main types of layers; convolutional, pooling and fully connected, each one of them playing a different role. Every layer of the CNN receives an input, transforms it and delivers an output. This way, the initial layers, which are convolutional ones, deliver feature maps out of the input images, whose complexity is reduced by the pooling layers. Eventually, these maps are led to the fully connected layers, where the feature maps are converted into vectors used for classification.

A typical CNN architecture is shown in Figure 2.

Some systems, like the one in [106], where a YoLOv3 CNN is used, take the input image and modify its scale to get several feature maps out of the same image. In this case, the CNN is used to generate three different sets of feature maps, based on three image scales, which eventually, after going through the fully connected layers, will be used for classification.

A similar approach is used in [97], where a YoLOv3 CNN identifies people. Identified people are tracked, and a CNN ANN extracts characteristic features from each person in the image. The feature vectors are passed to an LSTM ANN whose main task is to retain features over time so the temporal dimension can be added to the spatial features obtained by the CNN. The final feature vectors, coming out of the LSTM layers, are sent to a fully connected layer, which implements a Softmax algorithm used for event classification.

In [87], the object detection task, performed by a YoLO CNN, is combined with object tracking, a task developed by DeepSORT [129], a CNN architecture able to track multiple objects after they have been detected.

The approach made in [82] to detect a fallen person uses a YoLOv3 CNN to detect fallen bodies on the ground plane. It maximizes the sensitivity by turning 90 and 270 degrees all images and compare the bounding boxes found in the same image. Then, features are extracted from the bounding box, which will be used as classification features.

In [78,86], a wide residual network, which is a type of CNN, takes as input an OF and derives feature maps out of it. These maps are delivered to the fully connected layers, which, in turn, will pass vectors for movement classification to the last layers of the ANN.

A similar procedure is followed by the system in [89], whose ANN mixes layers of CNN, which deliver features maps from the incoming binarized video signal, with layers of radial basis function neural networks (RBFNN), which will be used as a classifier.

Another interesting type of CNN is the hourglass convolutional auto-encoder (HCAE), introduced in [103]. This kind of architecture piles convolutional and pooling layers over fully connected ones to get a feature vector, and then it follows the inverse process to reconstruct the input images. The HCAE compares the error value between the encoded-decoded frames and the original frames, applying back-propagation for self-tuning. Ten consecutive frames are inputted into the system to guarantee it captures both image and action features.

An alternate approach is the one presented in [66], where a CNN identifies objects (including people) and associate vectors to them. These vectors, which measure features, characterize both the human shape itself and its spatial relations with surrounding objects. This way, events are classified not only as a function of geometrical features of the silhouette but also as a function of its spatial relations with other objects present in the image. This approach has proven very useful to detect incomplete falls where pieces of furniture are involved.

A good number of approaches, as in [70], use 3D CNNs to extract spatiotemporal features out of 2D images, like the ones used in this system. This way, ANNs are used not only to extract spatial features associated with pose recognition but also to capture the temporal relation established among successive poses leading to a fall. The system in [52] uses this approach, creating a dynamic image by fusing in a single image all the frames belonging to a time window and passing this image to the ANN as the input from where extracting features.

Certain convolutional architectures, like the ones integrated into OpenPose and used in [87,90], can identify human body key points through convolutional pose machines (CPM), as shown in Figure 3, a CNN able to identify those features. These key points are used to build a vector model of the human body in a bottom-up approach.

To correct possible mistakes, this approximation is complemented in [90] by a top-down approach through single shot multibox detector-MobileNet (SSD-MobileNet), another convolutional architecture able to identify multiple objects, human bodies in this case. SSD-MobileNet, lighter and requires less computational power than typical SSDs, is used to remove all key points identified by OpenPose not being part of a human body, correcting this way, inappropriate body vector constructions.

A similar approach is used in [93], where a CNN is used to generate an inverted pendulum based on five human key points, knees, the center of the hip line, neck and head. The motion history of these joints is recorded, and a subsequent module calculates the pendulum rotation energy and its generalized force sequences. These features are then codified in a vector and used for classification purposes.

The system in [105] uses several ANNs and selects the most suitable one as a function of the environment and the characteristics of the tracked people. In addition, it uploads wrongly categorized images which are used to retrain the used models.

#### 4.2.3. Depth

Descriptors based on depth information have gained ground thanks to the development of low-cost depth sensors, such as Microsoft Kinect^®^. This affordable system counts with a software development kit (SDK) and applications able to detect and track joints and construct human body vector models. These elements, together with the depth information from stereoscopic scene observation, have raised great interest among the artificial vision research community in general and the human fall detection system developers in particular.

A good number of the studied systems use depth information, solely or together with RGB one, as the data source in the abstraction process leading to image descriptor construction. These systems have proved to be able to segment foreground, greatly diminishing interference due to illumination interferences up to the distance where stereoscopic vision procedures are able to infer depth data. Fall detection systems use this information either as depth maps or skeleton vector models.

##### Depth Map Representation

Depth maps, unlike RGB video signals, contain direct three-dimensional information on objects in the image. Therefore, depth map video signals integrate raw 3D information, so three-dimensional characterization features can be directly extracted from them.

This way, the system described in [46] identifies 16 regions of the human body marked with red tape and position them in space through stereoscopic techniques. Taking that information as a base, the system builds the body vector (aligned with spine orientation) and identifies its center of gravity (CG). Acceleration of CG and body vector angle on a vertical axis will be used as features for classification.

Foreground segmentation of human silhouette is made by these systems through depth information, by comparing depth data from images and a reference established at system startup. This way, pixels appearing in an image at a distance different from the one stored for that particular pixel in the reference are declared as foreground. This is the process followed by [44] to segment the human silhouette. In an ulterior step, descriptors based on bounding box, centroid, area and orientation of the silhouette are extracted.

Other systems, like the one in [101], extract background by using the same process and the silhouette is determined as the major connected body in the resulting image. Then, an ellipse is established around it, and classification will be made as a function of its aspect ratio and centroid position. A similar process is followed in [60], where, after background subtraction, an ellipse is established around the silhouette, and its centroid elevation and velocity, as well as its aspect ratio, are used as classification features.

The system in [57] uses depth maps to segment silhouettes as well and creates a bounding box around them. Box top coordinates are used to determine the head velocity profile during a fall event, and its Hausdorff distance to head trajectories recorded during real fall events is used to determine whether a fall has taken place. The Hausdorff distance quantifies how far two subsets of a metric space are from each other. The novelty of this system, leaving aside the introduction of the Hausdorff distance as described in [130], is the use of a moving capture (MoCap) technique to drive a human model using software to simulate its motion (OpenSim), so profiles of head vertical velocities can be captured in ADLs, and a database can be built. This database is used, by the introduction of the Hausdorff distance, to assess falls.

The system in [85], after foreground extraction by using depth information as in the previous systems, transforms the image to a black and white format and, after de-noising it through filtering, calculates the HOG. To do it, the system determines the gradient vector and its direction for each image pixel. Then, a histogram is constructed, which integrates all pixels’ information. This is the feature used for classification purposes.

In [42], silhouettes are tracked by using a proportional-integral-differential (PID) controller. A bounding box is created around the silhouette, and features are extracted in accordance with [131]. A fall will be called if thresholds established for features are exceeded. Faces are searched, and when identified, the tracking will be biased towards them.

Some other systems, like the one in [15], subtracts background by direct use of depth information contained in sequential images, so the difference between consecutive depth frames is used for segmentation. Then, the head is tracked, so the head vertical position/person height ratio can be determined, which, together with CG velocity, is used as a classification feature.

In [54], all background is set to a fixed depth distance. Then, a group of 2000 body pixels is randomly chosen, and for each of them, a vector of 2000 values, calculated as a function of the depth difference between pairs of points, is created. These pairs are determined by establishing 2000 pixel offset sets. The obtained 2000-value vector is used as a characteristic feature for pose classification.

The system introduced in [11], after the human silhouette is segmented by using depth information through a GMM process, calculates its curvature scale space (CSS) features by using the procedures described in [12]. CSS calculation method convolutes a parametric representation of a planar curve, silhouette edge in this case, with a Gaussian function. This way, a representation of the arc length vs. curvature is obtained. Then, silhouettes features are encoded, together with the Gaussian mixture model used in the aforementioned CSS process, in a single Fisher vector, which will be used, after being normalized, for classification purposes.

Finally, a block of systems creates volumes based on normal distributions constructed around point clouds. These distributions, called voxels, are grouped together, and descriptors are extracted out of voxel clusters to determine, first, whether they represent a human body and then to assess if it is in a fallen state.

This way, the system presented in [27] first estimates the ground plane by assuming that most of the pixels belonging to every horizontal line are part of the ground plane. The ground can then be estimated, line per line, attending to the pixel depth values as explained in the procedure described in [132]. To clean up the pictures, all pixels below the ground plane are discarded. Then, normal distributions transform (NDT) maps are created as a cloud of points surrounded by normal distributions with the physical appearance of an ellipsoid. These distributions, created around a minimum number of points, are called voxels and, in this system, are given fixed dimensions. Then, features that describe the local curvature and shape of the local neighborhood are extracted from the distributions. These features, known as IRON [133], allow voxel classification as being part of a human body or not and, this way, voxels tagged as human are clustered together. IRON features are then calculated for the cluster representing a human body, and the Mahalanobis distance between that vector and the distribution associated with fallen bodies is calculated. If the distance is below a threshold, the fall state is declared.

A similar process is used in [34], where, after the point cloud is truncated by removing all points not contained in the area in between the ground plane and a parallel one 0.7 m over it by applying the RANSAC procedure [134], NDTs are created and then segmented in patches of equal dimensions. A support vector machine (SVM) classifier determines which ones of those patches belong to a human body as a function of their geometric characteristics. Close patches tagged as humans are clustered, and a bounding box is created around. A second SVM determines whether clusters should be declared as a fallen person. This classification is refined, taking data from a database of obstacles of the area, so if the cluster is declared as a fallen person, but it is contained in the obstacle database, the declaration is skipped.

##### Skeleton Representation

Systems implementing this representation are able to detect and track joints and, based on that information, they can build a human body vector model. This block of techniques, as the previous one, strongly diminishes the noise associated with illumination but have problems to build a correct model when occlusion appears, both the one generated by obstacles and the one product of perspective auto-occlusions.

A good number of these systems are built over the Microsoft Kinect^®^ system and take advantage of both de SDK and the applications developed for it. This is the case of the system introduced in [40], where three Kinect^®^ systems cover the same area from different perspectives, and joints are, therefore, followed from different angles, reducing this way the tracking problems associated with occlusion. In this system, human movement is characterized through two main features, head speed and CG situation referenced to ankles position.

The Kinect^®^ system is also used in [65] to follow joints and estimate the vertical distance to the ground plane. Then, the angle between the vertical and the torso vector, which links the neck and spine base, is determined and used to identify a start keyframe (SKF), where a fall starts, and an end keyframe (EKF), where it ends. During this period, vertical distance to the ground plane and vertical velocity of followed upper joints will be the input for classification. A very similar approach is followed in [33], where torso/vertical angle and centroid height are the key features used for classification.

This system is used as well in [5] to build, around identified joints, both 2D and 3D bounding boxes aligned with the spine direction. Then, the ratio width/height is determined, and the relation H_CG_/P_CG_, being the former de elevation of the CG over the ground plane and the latter de distance between the CG projection on the ground and the support polygon defined by ankles position, is calculated. Those features will be the base for event classification.

In [135], human body key points are identified by a CNN whose input is a 2D RGB video signal complemented by depth information. Based on those key points, the system builds a human body vector model. A filter was developed to generate digital terrain models from data captured by airborne systems [136], and the depth data were then used to estimate the ground plane. The system uses all that information to calculate the distance from the body CG and the body region over the shoulders to the ground. These distances will serve to characterize the human pose.

A CNN is also used in [61] to generate feature maps out of the depth images. This network stacks convolution layers to extract features and pooling layers to reduce map complexity, with a philosophy identical to the one used in the RGB local characterization. The output map goes through two layers of fully connected layers to classify the recorded activity, and a Softmax function is implemented in the last layer of the ANN, which determines whether a fall has taken place.

In [84], prior to input images in a CNN to generate feature maps, which will be used for classification, the background is subtracted through an algorithm that combines depth maps and 2D images to enhance segmentation performance. This way, if the pixels of the segmented 2D silhouette experiment sharp changes, but pixels in the depth map do not, pixels subject to those changes are regarded as noise. The system mixes information from both sources, allowing a better track on segmented silhouettes and a quick track regain in case it is lost.

The system in [13]—after identifying human body joints as the key features whose trajectory will be used to determine whether a falling event has taken place—proposes rotating the torso so it is always vertical. This way, joint extraction becomes pose invariant, a technique used in the system with positive results in order to deal with the noise associated with joint identification as a result of rapid movement and occlusion, characteristic of falls.

### 4.3. Classification

Once pose/movement abstract descriptors have been extracted from video images, the next step of the fall detection process is classification. In broad terms, during this phase, the system classifies movement and or pose as a fall or a fallen state through an algorithm that is part of one of these two categories; generative or discriminative models.

Discriminative models are able to determine boundaries between classes, either by explicitly being given those boundaries or by setting them themselves using sets of pre-classified descriptors.

Generative models approach the classification problem in a totally different way, as they explicitly model the distribution of each class and then use the Bayes theorem to link descriptors to the most likely class, which, in this case, can only be a fall or a not fall state.

#### 4.3.1. Discriminative Models

The final goal of any classifier is assigning a class to a given set of descriptors. The discriminative models are able to establish the boundaries separating classes, so the probability of a descriptor belonging to a specific class can be given. In other terms, given α as a class, and [A] as the matrix of descriptor values associated with a pose or movement, this family of classifiers is able to determine the probability **P** (α|[A]).

##### Feature-Threshold-Based

Feature-threshold-based classification models are broadly used in the studied systems. This approach is easy and intuitive, as the researcher establishes threshold values for the descriptors, so their associated events can be assigned to a specific class in case those thresholds are exceeded.

This is the case of the system proposed in [31]. It classifies the action as a fall or a non-fall in accordance with a double rationale. On one hand, it establishes thresholds of ellipse features to estimate whether the pose fits a fallen state; on the other, an MHI feature exceeding a certain value indicates a fast movement and, therefore, a potential fall. The system proposed in [14] adds acceleration to the former features and, in [40], head speed over a certain threshold and CG position out of the segment defined by ankles are indicatives of a fall.

Similar approaches, where threshold values are determined by system developers based on previous experimentation, are implemented in a good number of the studied systems, as they are simple, intuitive and computationally inexpensive.

##### Multivariate Exponentially Weighted Moving Average

Multivariate exponentially weighted moving average (MEWMA) is a statistical process control to monitor variables that use the entire history of values of a set of variables. This technique allows designers to give a weighting value to all recorded variable outputs, so the most recent ones are given higher weight values, and the older ones are weighted lighter. This way, the last value is weighted λ (being λ a number between 0 and 1) and previous β values are weighted $\lambda^{\beta }$. Limits to the value of that weighted output are established, taking as a basis the expected mean and standard deviation of the process. Certain systems, like [28], use this technique for classification purposes. However, as it is unable to distinguish between falling events and other similar ones, events tagged as fall by the MEWMA classifier need to go through an ulterior support vector machine classifier.

##### Support Vector Machines

Support vector machines (SVM) are a set of supervised learning algorithms first introduced by Vapnik et al. [137].

SVMs are used for regression and classification problems. They create hyperplanes in high dimension spaces that separate classes nonlinearly. To fulfill this task, SVMs, similar to artificial neural networks, use kernel functions of different types.

A standard SVM boundary definition is shown in Figure 4.

In [74], linear, polynomial, and radial kernels are used to obtain the hyperplanes; in [67], radial ones are implemented, and in [48], polynomial kernels are used to achieve nonlinear classifications.

The support vector data description (SVDD), introduced by Tax et al. [138], is a classifying algorithm inspired by the support vector machine classifier, able to obtain a spherically shaped boundary around a dataset and, analogously to SVMs, it can use different kernel functions. The method is made robust against outliers in the training set and is capable of tightening classification by using negative examples. SVDDs classifying algorithms are used in [90].

SVMs have been very used in the studied systems as they have proofed to be very effective; however, they require high computational loads, something inappropriate for edge computing systems.

##### K-Nearest Neighbor

K-nearest neighbor (KNN) is an algorithm able to model the conditional probability of a sample belonging to a specific class. It is used for classification purposes in [16,17,48,74] among others.

KNNs assume that classification can be successfully made based on the class of the nearest neighbors. This way, if for a specific feature, all µ closest sample neighbors are part of a determined class, the probability of the sample being part of that class will be assessed as very high. This study is repeated for every feature contained in the descriptor, so a final assessment based on all features can be made. The algorithm usually gives different weights to the neighbors, and heavier weights are assigned to the closest ones. On top of that, it also assigns different weights to every feature. This way, the ones assessed as most relevant get heavier weights.

##### Decision Tree

Decision trees (DT) are algorithms used both in regression and classification. It is an intuitive tool to make decisions and explicitly represents decision-making. Classification DTs use categorical variables associated with classes. Trees are built by using leaves, which represent class labels, and branches, which represent characteristic features of those classes. DTs built process is iterative, with a selection of features correctly ordered to determine the split points that minimize a cost function that measures the computational requirements of the algorithm. These algorithms are prone to overfitting, as setting the correct number of branches per leaf is usually very challenging. To reduce the complexity of the trees, and therefore, their computational cost, branches are pruned when the relation cost-saving/accuracy loss is satisfactory. This type of classifier is used in [87,89].

Random forest (RF), like the one used in [54,87], is an aggregation technique of DT, introduced by Braiman [139], which main objective is avoiding overfitting. To accomplish this task, the training dataset is divided into subgroups, and therefore, a final number of DTs, equal to the number of dataset subgroups, is obtained. All of them are used in the process, so the final classification decision is actually a combination of the classification of all DTs.

Gradient boosting decision trees (GBDT) is another DT aggregation technique whose algorithm was first introduced by Friedman [140] where simple DTs are built and, for each one of them, a classification error in training time is determined. An error function based on calculated individual errors is determined, and its gradient is minimized by combining individual DT classifications in a proper way. This aggregation technique, specifically developed for DTs, is actually part of a broader family that will be more extensively presented in the next section.

Both techniques, RF and GBDT, are used in [87].

##### Boost Classifier

Boost classifier algorithms are a family of classifier building techniques that create strong classifiers by grouping weak ones. It is done by adding up models created from the training data until the system is perfectly predicted or a maximum number of models is reached.

This is done by building a model from the training data. Then, a second model is created to correct the errors from the first one. Models are added until the training set is well predicted or a maximum number of them is added. During the boosting process, the first model is trained on the entire database while the rest are fitted to the residuals of the previous ones.

Adaboost, used in [23], can be utilized to increase performances with any classification technique, but it is most commonly used with one-level decision trees.

In [64], boosting techniques are used on a J48 algorithm, a tree-based technique, similar to random forest, which is used to create univariate decision trees.

##### Sparse Representation Classifier

Sparse representations classification (SRC) is a technique used for image classification with a very good degree of performance.

Natural images are usually rich in texture and other structures that tend to be recurrent. For this reason, sparse representation can be successfully applied to image processing. This phenomenon is known as patch recurrence and, because of it, real-world digital images can be recognized by properly trained dictionaries.

SRCs are able to recognize those patches, as they can be expressed as a linear combination of a limited number of elements that are contained in the classifier dictionaries.

This is the case of the SRC presented in [24].

##### Logistic Regression

Logistic regression is a statistical model used for classification. It is able to implement a binary classifier, like the one needed to decide whether a fall event has taken place. For such a purpose, a logistic function is used. It can be adjusted by using classifying features associated with events tagged as fall or not fall.

This method is used in systems like [93], where a logistic classifying algorithm is employed to classify events as fall or not a fall, based on a vector that encodes the temporal series of rotation energy and generalized force.

Some artificial neural networks implement a logistic regression function for classification, like the one described in [106], where a CNN uses this function to determine the detection probability of each defined class.

##### Deep Learning Models

In [83], the last layers of the ANN implement a Softmax function, a generalization of the logistic function used for multinomial logistic regression. This function is used as the activation function of the nodes of the last layer of a neural network, so its output is normalized to a probability distribution over the different output classes. Softmax is also implemented in the last layers of the artificial neural networks used in [75,103], among other studied systems.

Multilayer perceptron (MLP) is a type of multilayered ANN with hidden layers between the entrance and the exit ones able to sort out classes non linearly separable. Each node of this network is a neuron that uses a nonlinear activation function, and it is used for classification purposes in [48,87].

Radial basis function neural networks (RBFNN) are used in the last layer of [89] to classify the feature vectors coming from previous CNN layers. This ANN is characterized by using radial basis functions as activation functions and yields better generalization capabilities than other architectures, such as Softmax, as it is trained via minimizing the generalized error estimated by a localized-generalization error model (L-GEM).

Often, the last layers of ANN architectures are fully connected ones, as in [58,76,86], where all nodes of a layer are connected to all nodes in the next one. In these structures, the input layer is used to flatten outputs from previous layers and transform them into a single vector, while subsequent layers apply weights to determine a proper tagging and, therefore, successfully classify events.

Finally, another ANN structure useful for classification is the autoencoder one, used in [70]. Autoencoders are ANNs trained to generate outputs equal to inputs. Its internal structure includes a hidden layer where all neurons are connected to every input and output node. This way, autoencoders get high dimensional vectors and encode their features. Then, these features are decoded back. As the number of dimensions of the output vector may be reduced, this kind of ANNs can be used for classification purposes by reducing the number of output dimensions to the number of final expected classes.

#### 4.3.2. Generative Models

The approach of generative models to the classification problem is completely different from the one followed by the discriminative ones.

Generative models explicitly model the distribution of each class. This way, given α as a class, and [A] as the matrix of descriptor values associated with a pose or movement, if both P ([A]|α) and P (α) can be determined, it will be possible, by direct application of the Bayes theorem, to obtain P (α|[A]), which will solve the classification problem.

##### Hidden Markov Model

Classification using the hidden Markov model (HMM) algorithm is one of the three typical problems that can be solved through this procedure. It was first proposed with this purpose by Rabiner et al. [141] to solve the speech recognition problem, and it is used in [100] to classify the feature vectors associated with a silhouette.

HMMs are stochastic models used to represent systems whose state variables change randomly over time. Unlike other statistical procedures, like Markov chains, which deal with fully observable systems, HMMs tackle partially observable systems. This way, the final objective of the HMM classifying problem resolution will be decided, on the basis of the observable data (feature vector), whether a fall has occurred (hidden system state).

The system proposed in [100] determines, using an HMM as a classifier, on the basis of silhouette surface, centroid position and bounding box aspect ratio, whether a fall takes place or not. To do it, and to take as a reference recorded falls, a probability is assigned to the two possible system states (fall/not fall) based on value and variation along the event timeframe period of the feature vector. This classifying technique is used with success in this system, though in [142], a brief summary of the numerous limitations of this basic HMM approach is presented, and several more efficient extensions of the algorithm, such as variable transition HMM or the hidden semi-Markov model, are introduced. These algorithm variations are developed as the basic HMM process is considered ill-suited for modeling systems where interacting elements are represented through a vector of single state variables.

A similar classification approach using an HMM classifier is used in [47], where future states predicted by an autoregressive-moving-average (ARMA) algorithm are classified as fall or not-fall events. ARMA models are able to predict future states of a system based on a previous time-series. The model integrates two modules, an autoregressive one, which uses a linear combination of weighted previous system state values, and a moving average one, which linearly combines weighted previous errors between system state real values and predicted ones. In the model, errors are assumed to be random values that fit a Gaussian distribution of mean 0 and variance σ^2^.

### 4.4. Tracking

A good number of the reviewed systems identify objects through ANN or extract silhouettes from the background. Then, relevant features are associated with the already segmented objects. This assignment requires a constant update, and, therefore, object correlation needs to be established from frame-to-frame. This correlation is made through object tracking, and a good number of different techniques are used for such a purpose.

#### 4.4.1. Moving Average Filter

The double moving average filter used in [65] smooths vertical distance from joints to the ground plane. This filter determines twice the mean value of the last n samples, acting this way as a low pass filter, eliminating high-frequency signal components associated with noise.

#### 4.4.2. PID Filter

The system proposed in [42] uses a proportional-integral-differential (PID) filter to maintain tracking on silhouettes segmented from the background. Constants of the filter to guarantee smooth tracking, reducing overshoots and steady-state errors, are calculated through a genetic algorithm. This algorithm, inspired by the theory of natural evolution, is a heuristic search where sets of values are selected or discarded based on its ability to reduce to a minimum the absolute error function and, therefore, minimize overshoots and steady errors.

#### 4.4.3. Kalman Filter

Kalman filter, first introduced by R. E. Kalman in [143], is a recursive algorithm that allows improvements in the determination of system variable values by combining several sets of indirect system variable observations containing inaccuracies. The resulting estimation is more precise than any of the ones which could be inferred from a single indirect observation set.

This way, in [40], the tracking of joints, followed by three independent Kinect^®^ systems, is fused by a Kalman filter. The resulting joint position is estimated by integrating information from the three systems and is more accurate than one of any of the individual systems.

A particular variation in the use of Kalman filtering is the one in [97], where a procedure call deep-sort, presented in [129], is used. In this process, a Kalman algorithm is used to estimate the next location of the tracked person, and then the Mahalanobis distance is calculated between the detected person in the following frame and its estimated position. By measuring this distance, uncertainty in the track correlation can be quantified. This filter performance is deeply affected by occlusion. To mitigate this problem, the uncertainty value is associated with the track descriptor and, to keep tracks after long occlusion periods, the process saves those descriptors for 100 frames.

Although this filtering algorithm works very well to maintain tracks in linear systems, human bodies involved in a fall tend to behave nonlinearly, substantially degrading its ability to maintain tracking.

#### 4.4.4. Particle Filter

This method, used in [15], is a Monte Carlo algorithm used for object tracking in video signals. Introduced in 1993 by Gordon [144] as a Bayesian recursive filter, it is able to determine future system states, in this case, future positions of the tracked object.

The filter algorithm follows an iterative approach. This way, after a cloud of particles, image pixels, in this case, have been selected, weights are assigned to them. Those weigh values are a function of the probability of being part of the tracked object. Then, the initial particle cloud is updated by using the weight values. Based on object cinematic, its movement is propagated to the particle cloud, predicting, this way, the future object situation. The process continues with a new update phase to guarantee the predicted cloud matches the tracked object.

This algorithm, although affected by occlusion, has proven to be highly capable of maintaining tracks on objects moving nonlinearly and, therefore, the result is adequate to track human bodies during fall events.

Rao–Blackwellized particle filter (RBPF), like the one used in [63], is a type of particle filter tracking algorithm used in linear/nonlinear scenarios where a purely Gaussian approach is inadequate.

This algorithm divides particles into two sets. Those which can be analytically evaluated and those which cannot. This way, the filtering equations are separated into two sets, so two different approaches can be used to calculate them. The first set, which includes linear moving particles, is solved by using a Kalman filter approach, while the second one, whose particles move nonlinearly, is solved by employing a Monte Carlo sampling method.

#### 4.4.5. Fused Images

In [9], a fusing center fuses images taken from orthogonal views, and the obtained object is tagged with a number. Objects identified in the next frame are correlated to previous ones if they meet the minimum distance established threshold. This way, the tracking is maintained.

#### 4.4.6. Camshift

This algorithm, integrated into OpenCV and used in [59], first converts images RGB to hue-saturation-value (HSV) and, starting with frames where a CNN has created a bounding box (BB) around a detected person, it determines the hue histogram in each BB. Then, morphological operations are applied to reduce noise associated with illumination. In the consecutive frame, the area which better fits the recorded Hue histogram is established and compared with detected BBs. That way, a correlation can be established and, therefore, a track on a person.

#### 4.4.7. Deep Learning Architectures

DeepSORT is a CNN used to track multiple objects at the same time, as shown in [87].

The system presented in [71] tracks images using an algorithm as follows: First, in every new frame, a YoLO convolutional architecture is used to identify people. Once all people in the frame have been identified, a Siamese CNN is used to first determine the characteristic features of every person identified in the frame and then compare them with the ones associated with people identified in previous frames, looking for similarities. At the same time, an LSTM ANN is used to predict people’s motion, so associations to maintain track of people from frame-to-frame can be made. Based on feature similarity and movement association, a track can be established on people present in consecutive video frames or can be started when a new person appears for the first time in a video sequence. An almost equal process is used in [97] to keep track of people with two CNNs working in parallel, a first one to identify people and a second one to extract characteristic features out of them. That way, tracks can be established.

In [41], a CNN is used to detect people in every frame. A BB is established around, and distances from central point BBs of consecutive frames are determined. Boxes meeting minimum distance criteria in consecutive frames are correlated and, this way, tracking is established.

### 4.5. Classifying Algorithms Performances

A number of the reviewed systems establish comparisons with other ones. Many of them base that comparison on performance figures obtained on different datasets, while some others establish a system-to-system comparison based on the same database. However, systems are, in broad terms, an aggregation of two main blocks, the first one whose mission is inferring descriptors from images and a second one that classifies those features. This way, system comparison, even on the same dataset, compares two aggregated blocks so, comparisons on performances of a specific block is difficult to assess, as it is influenced by the other one.

To avoid these problems, these comparisons have been ignored. The only ones taken into consideration have been those that compare one of the blocks and are based on the same dataset. The results are shown in Table 2. In global terms, SVMs and deep learning classifiers are the ones with better performances. The best working classifying deep learning architectures are MLP, autoencoders and those implementing Softmax algorithms like GoogLeNet. It is also relevant that in accordance with C.J. Chong et al. [3], systems whose descriptors are dynamic and, therefore, include references to the time variable, have better performances than those other ones whose descriptors do not incorporate that variable.

### 4.6. Validation Datasets

The systems included in this research have been tested by using datasets. On many occasions, those datasets have been specifically developed by the researchers to test and validate their systems, so their performances can be determined. These datasets, although briefly discussed in the articles presenting the systems, are not usually publicly accessible.

However, there are also a group of datasets used in the system validation and performance determination phases that are public. Most of them are also accessible through the Internet, so developers can download and use them for research purposes. All the datasets belonging to this category used in the development of the systems contained in this review are collected in Table 3.

Datasets associated with the reviewed systems, both the publicly accessible ones and the ones that are not, are recorded either by volunteers or actors young and fit enough to guarantee that a simulated fall will not harm them. In some of them, actors are advised by therapists, so they can imitate how an elderly person moves or falls. Finally, none of the databases include elderly real falls or daily life activities performed by elderly people.

The datasets are grouped by collected signal type, so five big groups are identified.

The first group is integrated by a single dataset. It collects falls and activities of daily life (ADL) executed by volunteers whose results are recorded using different sensors, included RGB and IR cameras. It is used by a single system for validation purposes;The second group, which includes three datasets, incorporates depth and accelerometric data. By its relevance and number of reviewed systems using it in their performance evaluation, one dataset is especially important, UR fall detection [29]. This dataset, employed by over a third of all studied systems, includes 30 falls and 40 ADLs recorded by two depth systems, one providing frontal images and a second camera recording vertical ones. This information is accompanied by accelerometric data and was released in 2015;The third group is composed of nine datasets. They all mix ADLs and falls recorded in different scenarios by RGB cameras, either conventional or fish eye ones, from different perspectives and at different heights. Two of them exceed the mark of 10% users, LE2I [23] and the Multicam Fall Dataset [10].LE2I, published in 2013, is a dataset that includes 143 different types of falls performed by actors and 48 ADLs. These events were recorded in environments simulating the ones that could be found in an elderly home.Multicam includes 24 scenarios recorded with 8 IP cameras, so events can be analyzed from multiple perspectives. Twenty-two of the scenarios contains falls, while the other 2 only include confounding actions. Events are simulated by volunteers, and this dataset was released in 2010;The fourth group includes six datasets. Different activities, falls included, are recorded by depth systems. The two most used ones are the Fall Detection Dataset [30] and SDUFall [12], though both of them fall below the 10% users mark.Fall Detection Dataset, used by almost 10% of the systems, was published in 2017. The images in this dataset are recorded in five different rooms from eight different view angles, and five different volunteers take part in it.SDUFall, published in 2014, is another dataset that gathers depth information associated with six types of actions, being a fall one of them. Actions are repeated 30 times by 10 volunteers and are recorded by a depth system;The fifth group, composed of a single dataset, collects synthetic information. CMU Graphics Lab—motion capture library [55] is a dataset that contains biomechanical information related to human body movement captured through the use of motion capture (MoCap) technology. To generate that information, a group of volunteers, wearing sensors in different parts of their bodies, execute diverse activities. The information collected by the sensors is integrated through a human body model and stored in the dataset, so it can be used for development purposes. This approach to system development and validation has numerous advantages over conventional methods, as it gives developers the possibility of training their systems under any possible image perspective or occlusion situation. However, clutter and noise, the other important problems for optimal system performance, are not included in the information recorded in this database.

## 5. Conclusions

In a world with an aging population, where the number of people over 60 will soon over exceed the number of teenagers and youngsters below 24, the attention to elderly care will become an area of increasing relevance, where a growing amount of resources will be poured.

A good number of these resources will be used to automate some of the assistance tasks to the elderly community, and one of them will be unmanned person status surveillance, so an automated quick response can be triggered in case a dependent person goes through a distress situation.

One of those situations is accidental falls; as for the elderly community, over one-third of falls lead to major injuries, including, directly or indirectly, death. With that background scenario, automatic fall detection systems could be assessed as an area of growing interest over the course of the next few years, as they could have a high impact on life quality for the dependent community.

Among all potential technologies able to detect a fall, the artificial vision techniques have proven extremely effective over the last years. With the final goal of shedding light on the current state of research in that area, this review has been elaborated, so it can give a global picture of the techniques used to detect a fall through artificial vision systems to all new researchers interested in this field trying to decide how to start a new system design. This study intends to show the advantages and disadvantages of all user processes in an attempt to orientate new developers in a field that could contribute to reducing both dependency and care costs in the elderly community.

The systems based on artificial vision have deeply evolved over the course of the last five years. To determine the characteristics of this evolution, a thorough review of published information has been made, which has taken into consideration most of the literature published on vision-based fall detection research from 2015 to 2020.

These systems examine human pose, human movement or a mix of both and categorize them as fall in case the established criteria are met. All of them have a common structure of two blocks, a first one which assigns abstract descriptors to input video signals, and a second one which classifies them. In some of the reviewed systems, these two blocks are preceded by another one whose objective is improving the quality of the incoming signal by reducing noise or adapting its format to the needs of the blocks downstream it.

Almost all reviewed systems work either with RGB or depth video inputs. Systems working with RGB video signals have evolved from the use of global descriptors to the use of local ones. Global descriptors extract information from the foreground, once it has segmented, and encode it as a whole, while local ones focus on area patches from where relevant features, characteristic of human movement or pose, can be derived. This evolution has made systems more resilient to perspective changes and noise due to illumination and occlusion.

Depth information is also used either solely or complementing RGB images. The systems using it have proofed to be very reliable in high noise conditions due to illumination. However, higher prices and an effectiveness limitation up to distances where depth data can be inferred from stereoscopic vision remain relevant limitations to this technology.

The second block of these systems approaches the classifying problem from two possible perspectives, discriminative or generative. Discriminative models establish boundaries between classes, while generative ones model each class probability distribution.

Although an extensive array of techniques has been used to implement both blocks, the use of ANNs is becoming increasingly popular, as their ability to learn to give them a matchless advantage. This is the case of [105], a system that uses images that have raised false alarms for retraining. Among all possible ANN architectures, two families have proven to offer good performances in the field of artificial vision, convolutional (CNN) and recurrent ones (RNN). Convolutional networks are able to create feature maps out of images that express what can be seen in them. Recurrent architectures, and specially LSTMs, are able to grasp the dynamics associated with video clips, as the cycles in their structure allow them to remember passed features and link them along time. New architectures fusing layers of both networks, CNNs and LSTMs, being able to identify objects and abstract their movement, show promising results in the area.

After object identification, movement capture is needed, so its dynamics can be abstracted. To do it, object tracking is required. This activity can be done through a number of techniques that can be grouped into two blocks, linear and nonlinear. Due to the nonlinear nature of the movement of the human body during falls, the last block of techniques has proven to be more suitable for this purpose.

A number of datasets are used for system validation and performance determination purposes. However, their fragmentation and the total absence of a common reference framework for system performance evaluation make comparison very difficult. In addition, all datasets are recorded by actors or volunteers clearly younger than the elderly community. The significant differences between simulated and real falls and between falls of elderly and young people are documented by Kangas [145], and Klenk [146], so reasonable doubts on the performances of all reviewed systems in the real world are raised. In any case, the clash between privacy protection and real-world datasets makes it difficult to get good quality data for system training and validation.

No articles mentioning the orientation of system design towards their potential users have been found during this research. The only articles found in the area of fall detection systems regarding this aspect are the ones of Thilo et al. [147], and Demiris et al. [148], where the elderly community needs are described, and recommendations to developers are given. This way, there is evidence of a disconnection between developers and users, which, eventually, leads to no use of these systems.

The implementation of vision-based fall detection systems has traditionally fallen in the field of ambient systems. However, robots are offering the possibility of making them mobile, and the potential future incorporation of smart glasses or contacts gives the chance to make this system wearable. In these cases, cloud computing may not be an option, so the computational cost will need to be taken into consideration, and low-power consumption will be a key factor.

Finally, although this review has been solely focused on pure vision-based fall detection systems to diminish its extension, in accordance with L. Ren et al. [149], optimal detection performance comes from fusion-based systems that complement vision-based technologies with alternative ones.

## Figures and Tables

**Figure 1 sensors-21-00947-f001:**
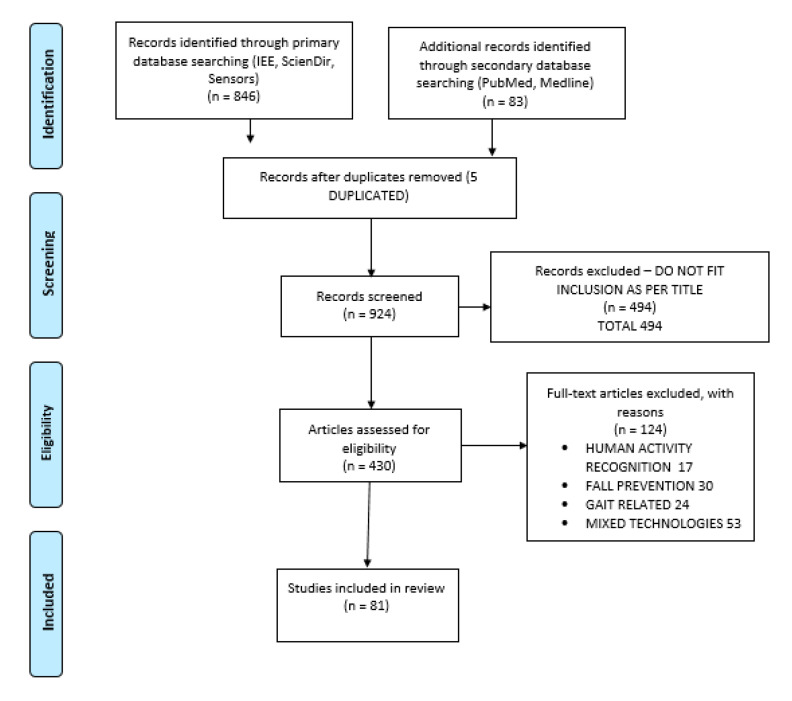
Flow diagram of adopted search and selection strategy for paper selection.

**Figure 2 sensors-21-00947-f002:**
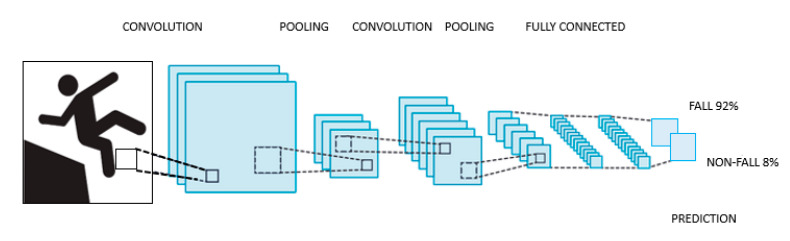
Typical convolutional neural network (CNN) architecture.

**Figure 3 sensors-21-00947-f003:**
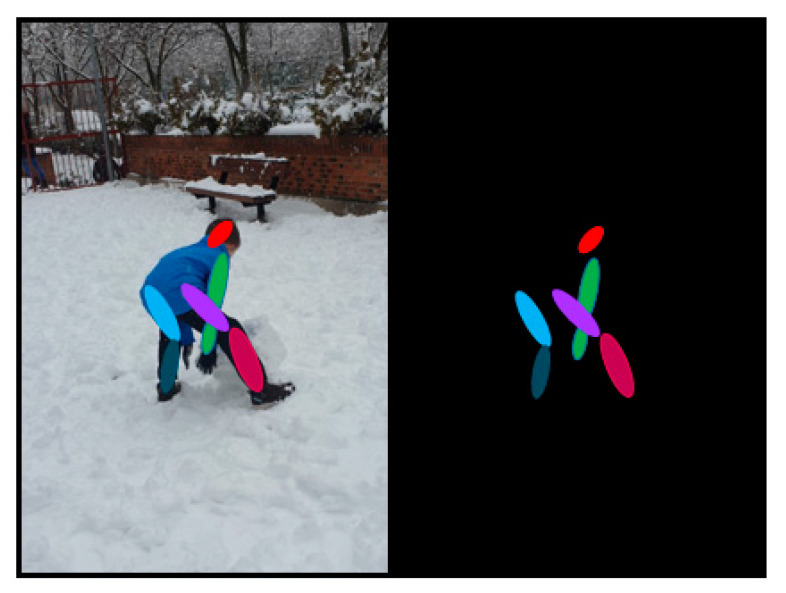
Convolutional pose machine presentation.

**Figure 4 sensors-21-00947-f004:**
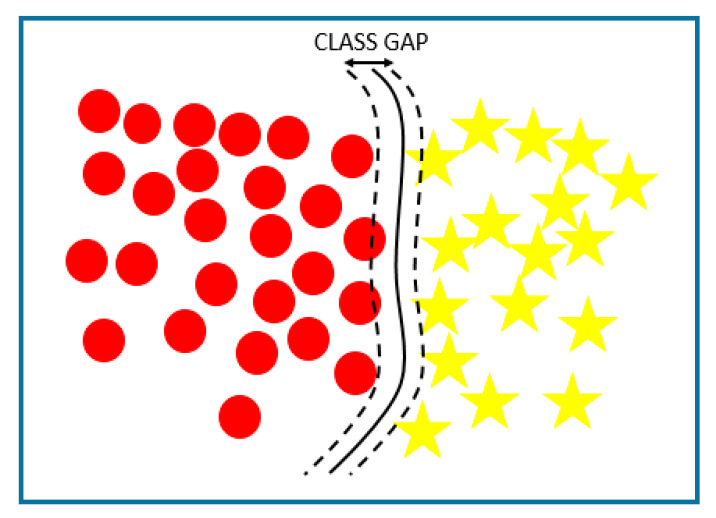
Support vector machine boundary definition.

**Table 1 sensors-21-00947-t001:** Vision-based fall detection systems published 2015–2020.

Reference	Year	Characterization (Global/Local/Depth)	Classification	Input Signal	Used Datasets	Performance
A. Yajai et al. [5]	2015	Skeleton joint tracking model provided by MS Kinect^®^ is used to track joints and build a 2D and 3D bounding box around the body/depth characterization	Feature-threshold-based. Height/width ratio of the bounding box center of gravity (CG) position in relation to support polygon (defined by ankle joints)	Depth	This system-specific video dataset—no public access at revision time	Accuracy 98.43% Specificity 98.75% Recall 98.12%
C. -J. Chong et al. [6]	2015	Pixel clustering and background (Horprasert)/global characterization	Feature-threshold-based. Method 1: Bounding box (BB) aspect ratio CG position Method 2: Ellipse orientation and aspect ratio Motion history image (MHI)	Red-green-blue (RGB)	Specific video dataset—no public access at revision time	Method 1 Sensitivity 66.7% Specificity 80% Method 2 Sensitivity 72.2% Specificity 90%
H. Rajabi et al. [7]	2015	Foreground extraction through background subtraction (Gaussian mixed models—GMM) and Sobel filter application/ global characterization	Feature-threshold-based. BB orientation angle Change of CG width Height/width relation of contour Hu moment invariants	RGB	This system-specific video dataset—no public access at revision time	Fall detection success rate 81%
L. H. Juang et al. [8]	2015	Foreground extraction through background subtraction (optical flow-based) and human joints identified/global characterization	Support vector machine (SVM)	RGB	This system-specific video dataset—no public access at revision time	Accuracy up to 100%
M. A. Mousse et al. [9]	2015	Foreground extraction through pixel color and brightness distortion determination and integration of foreground maps through homography/global characterization	Feature-threshold-based. Ratio observed silhouette area/silhouette area projected on the ground plane	RGB—2 ORTHOGONAL VIEWS	Multicam Fall Dataset [10]	Sensitivity 95.8% Specificity 100%
Muzaffer Aslan et al. [11]	2015	Human silhouette is segmented using depth information, and curvature scale space (CSS) is calculated and encoded in a Fisher vector/depth characterization	SVM	Depth	SDUFall [12]	Average accuracy 88.01%
Z. Bian et al. [13]	2015	Silhouette extraction by using depth information. Human body joints identified and tracked with torso rotation/depth characterization	SVM	Depth	This system-specific video dataset—no public access at revision time	Sensitivity 95.8% Specificity 100%
C. Lin et al. [14]	2016	Foreground extraction through background subtraction (GMM)/global characterization	Feature-threshold-based. Ellipse orientation Linear and angular acceleration MHI	RGB	This system-specific video dataset—no public access at revision time	Not published
F. Merrouche et al. [15]	2016	Foreground extraction by using the difference between depth frames and head tracking through particle filter/depth characterization	Feature-threshold-based. Ratio head vertical position/person height CG velocity	Depth	SDUFall [12]	Sensitivity 90.76% Specificity 93.52% Accuracy 92.98%
K. G. Gunale et al. [16]	2016	Foreground extraction through background subtraction (direct comparison)/global characterization	K-nearest neighbor (KNN)	RGB	Chute dataset—no public access at revision time	Accuracy Fall 90% No fall 100%
K. R. Bhavya et al. [17]	2016	Foreground extraction through background subtraction (direct comparison)/global characterization + optical flow (OF)/global characterization	KNN on MHI and OF features	RGB	This system-specific video dataset—no public access at revision time	Not published
Kun Wang et al. [18]	2016	Segmentation through vibe [19] and histogram of oriented gradients (HOG) and local binary pattern (LBP)/global characterization + feature maps obtained through convolutional neural network (CNN)/ local characterization	SVM-linear kernel	RGB	Multicam Fall Dataset [10] and SIMPLE Fall Detection Dataset [20] and This system-specific video dataset—no public access at revision time	Sensitivity 93.7% Specificity 92%
U. Pratap et al. [21]	2016	Foreground extraction through background subtraction (GMM)/global characterization	Feature-threshold-based. Silhouette CG stationary over a threshold time limit	RGB	Specific video datasets—no public access at revision time	Fall detection rate 92% False alarm rate 6.25%
X. Wang et al. [22]	2016	Segmentation through vibe [19] and upper body database populated and sparse OF determined/global characterization	Feature-threshold-based. Body ratio width/height Vertical velocity derived from OF Upper body position history	RGB	LE2I [23]	Average precision 81.55%
A. Y. Alaoui et al. [24]	2017	Foreground extraction through background subtraction (direct comparison)/global characterization + OF/global characterization	No classification algorithm reported	RGB	CHARFI2012 Dataset [25]	Precision 91% Sensitivity 86.66%
Apichet Yajai et al. [26]	2017	Skeleton joint tracking model provided by MS Kinect^®^/depth characterization	Feature-threshold-based. Aspect ratios: Bounding box CoG Bounding box diagonal vs. max. height Bounding box height vs. max. height	Depth	This system-specific video dataset—no public access at revision time	Accuracy 98.15% Sensitivity 97.75% Specificity 98.25%
B. Lewandowski et al. [27]	2017	voxels around the point cloud are calculated. The ones classified as human are clustered, and IRON features are calculated/local characterization	Feature-threshold-based. Mahalanobis distance between cluster IRON features and the distribution of IRON features from fallen bodies	Depth	This system-specific video dataset—no public access at revision time	Sensitivity in operational environments 99%
F. Harrou et al. [28]	2017	Foreground extraction through background subtraction (direct comparison)/depth characterization	Multivariate exponentially weighted moving average (MEWMA)-SVM KNN Artificial neural network (ANN) Naïve Bayes (NB)	RGB	UR Fall Detection [29] & Fall Detection Dataset [30]	Accuracy KNN 91.94% ANN 95.15% NB 93.55% NEWMA-SVM 96.66%
G. M. Basavaraj et al. [31]	2017	Foreground extraction through background subtraction (median)/global characterization	Feature-threshold-based. Ellipse eccentricity and orientation MHI	RGB	This system-specific video dataset—no public access at revision time	Accuracy Fall 86.66% Non-fall 90%
K. Adhikari et al. [30]	2017	Foreground extraction through background subtraction (direct comparison) using both RGB techniques and depth ones and Feature maps obtained through CNN/local and depth characterization	Softmax based on features vector from CNN	Depth	This system-specific video dataset—no public access at revision time	Overall, accuracy 74% System sensitivity to lying pose 99%
Koldo De Miguel et al. [32]	2017	Foreground extraction through background subtraction (GMM) + Sparse OF determined/global characterization	KNN on silhouette and OF features	RGB	This system-specific video dataset—no public access at revision time	Accuracy 96.9% Sensitivity 96% Specificity 97.6%
Leiyue Yao et al. [33]	2017	Skeleton joint tracking model provided by MS Kinect^®^/depth characterization	Feature-threshold-based Torso angle Centroid height	Depth	This system-specific video dataset—no public access at revision time	Accuracy 97.5% True positive rate 98% True negative rate 97%
M. Antonello et al. [34]	2017	voxels around the point cloud are calculated. Then they are segmented in homogeneous patches and the ones classified as human are gathered and classified or not as a human lying body/depth characterization	SVM—radial-based kernel	Depth	IASLAB-RGBD fallen person Dataset [35]	Set A Accuracy: single view (SV) 0.87/SV+map verification (MV) 0.92 Precision: SV 0.73/SV+MV 0.85 Recall: SV 0.85/SV+MV 0.85 Set B Accuracy: SV 0.88/SV+MV 0.9 Precision: SV 0.8/SV+MV 0.87 Recall: SV 0.86/SV+MV 0.81
M. N. H. Mohd et al. [36]	2017	Skeleton joint tracking model provided by MS Kinect^®^ is used to determine joint positions and speeds/depth characterization	SVM based on joints speeds and rule-based decision-based on joints position in relation to knees	Depth	TST Fall Detection [37] and UR Fall Detection [29] and Falling Detection [38]	Accuracy 97.39% Specificity 96.61% Sensitivity 100%
N. B. Joshi et al. [39]	2017	Foreground extraction through background subtraction (GMM)/global characterization	Feature-threshold-based. BB width/height ratio CG position Orientation Hu moments	RGB	LE2I [23]	Specificity 92.98% Accuracy 91.89%
N. Otanasap et al. [40]	2017	Skeleton joint tracking model provided by MS Kinect^®^/depth characterization	Feature-threshold-based. Head velocity CG position in relation to ankle joints	Depth	This system-specific video dataset—no public access at revision time	Sensitivity 97% Accuracy 100%
Q. Feng et al. [41]	2017	CNN is used to detect and track people, and Sub-MHI are correlated to each person BB/local characterization	SVM	RGB	UR Fall Detection [29]	Precision 96.8% Recall 98.1% F_1_ 97.4%
S. Hernandez-Mendez et al. [42]	2017	Foreground extraction through background subtraction (direct comparison) and silhouette tracking. Then centroid and features are determined/depth characterization	Feature-threshold-based. Angles and ratio height/width of the BB	Depth	Depth And Accelerometric Dataset [43] and this system-specific video dataset—no public access at revision time	The fallen pose is detected correctly on 100% of occasions.
S. Kasturi et al. [44]	2017	Foreground extraction through background subtraction (direct comparison)/depth characterization	SVM	Depth	UR Fall Detection [29]	Sensitivity 100% Specificity 88.33%
S. Kasturi et al. [45]	2017	Foreground extraction through background subtraction (direct comparison)/depth characterization	SVM	Depth	UR Fall Detection [29]	Accuracy Total testing accuracy 96.34%
S. Pattamaset et al. [46]	2017	Body vector construction and CG identification taking as starting point 16 parts of the human body/depth characterization	Feature-threshold-based. CG acceleration Body vector/vertical angle	Depth	This system-specific video dataset—no public access at revision time	Accuracy 100%
Sajjad Taghvaei et al. [47]	2017	Foreground extraction through background subtraction/depth characterization	Hidden Markov model (HMM)	Depth	This system-specific video dataset—no public access at revision time	Accuracy 84.72%
Y. M. Galvão et al. [48]	2017	Median square error (MSE) every 3 frames/global characterization	Multilayer perceptron (MLP) KNN SVM—polynomial kernel	RGB	UR Fall Detection [29]	F1 score: MLP 0.991 KNN 0.988 SVM—polynomial kernel 0.988
Thanh-Hai Tran et al. [49]	2017	Skeleton joint tracking model provided by MS Kinect^®^/depth characterization or Motion map extraction from RGB images and gradient kernel descriptor calculated/global characterization	Feature-threshold-based. Height of hip joint Vertical body velocity Or SVM classification	Depth or RGB	UR Fall Detection [29] and LE2I [23] and Multimodal Multiview Dataset of Human Activities [50]	UR Dataset Sensitivity 100% Specificity 99.23% LE2I Dataset Sensitivity 97.95% Specificity 97.87% MULTIMODAL Dataset (Average) Sensitivity 92.62% Specificity 100%
X. Li et al. [51]	2017	Foreground extraction through background subtraction (direct comparison) and feature maps obtained through CNN/ local characterization	Softmax based on features vector from CNN	RGB	UR Fall Detection [29]	Sensitivity 100% Specificity 99.98% Accuracy 99.98%
Yaxiang Fan et al. [52]	2017	Feature maps obtained through CNN from dynamic images/local characterization	Classification made by fully connected last layers of CNNs	RGB	Multicam Fall Dataset [10] & LE2I [23] and High-Quality Dataset [53] and This system-specific video dataset—no public access at revision time	Sensitivity LE2I 98.43% Multicam 97.1% HIGH-QUALITY FALL SIM 74.2% SYSTEM Dataset 63.7%
A. Abobakr et al. [54]	2018	Silhouette extraction by using depth information. A feature vector of different body pixels based on depth difference between pairs of points is created/depth characterization	Random decision forest for pose recognition and SVM for movement identification	Depth	UR Fall Detection [29] and CMU Graphics Lab—motion capture library [55]	Accuracy 96% Precision 91% Sensitivity 100% Specificity 93%
B. Dai et al. [56]	2018	Foreground extraction through background subtraction (direct comparison)/global characterization	Feature-threshold-based. BB segmented areas occupancy. CG/height ratio CG vertical speed	RGB	UR Fall Detection [29] and This system-specific video dataset—no public access at revision time	Sensitivity 95% Specificity 96.7%
Georgios Mastorakis et al. [57]	2018	Depth images are used to determine head velocity profile/depth characterization	Feature-threshold-based. Hausdorff distance between real head velocity profile and database ones	Depth	Specific video dataset developed for [43] (A) and [12] (B)– no public access at revision time	A Dataset Sensitivity 100% Specificity 100% B Dataset Sensitivity 90.88% Specificity 98.48%
K. Sehairi et al. [58]	2018	Foreground extraction through background subtraction (self-organizing maps) and feature extraction associated with each silhouette/global characterization	SVM-radial basis function (SVM-RBF) KNN Fully connected ANN trained through background propagation ANN	RGB	LE2I [23]	Accuracy SVM-RBF 99.27% KNN 98.91% ANN 99.61%
Kun-Lin Lu et al. [59]	2018	Person detection through CNN YoLOv3 and feature extraction of the generated bounding box/local characterization	Feature-threshold-based Bounding box height evolution in 1.5 s periods	RGB	This system-specific video dataset—no public access at revision time	Recall 100% Precision 93.94% Accuracy 95.96%
Leila Panahi et al. [60]	2018	Foreground extraction through background subtraction (depth information) and silhouette tracking. Then ellipse is established around the silhouette, and features are determined/depth characterization	SVM & Threshold-based decision Centroid elevation Centroid speed Ellipse aspect ratio	Depth	Depth and Accelerometric Dataset [43]	Average results SVM Sensitivity 98.52% Specificity 97.35% Threshold-based decision Sensitivity 98.52% Specificity 97.35%
M. Rahnemoonfar et al. [61]	2018	Feature maps obtained through CNN/depth characterization	Softmax based on features vector from CNN	Depth	SDUFall [12]	Accuracy 97.58%
Manola Ricciuti et al. [62]	2018	Foreground extraction through background subtraction (direct comparison)/depth characterization	SVM	Depth	This system-specific video dataset—no public access at revision time	Accuracy 98.6%
Myeongseob Ko et al. [63]	2018	Depth map from monocular images and silhouette detection through particle swarm optimization/global characterization	Feature-threshold-based Vertical velocity BB aspect ratio BB height Top depth/bottom depth ratio	RGB	This system-specific video dataset—no public access at revision time	Accuracy 97.7% Sensitivity 95.7% Specificity 98.7%
Syed F. Ali et al. [64]	2018	Foreground extraction through background subtraction (GMM)/global characterization	Boosted J48	RGB	UR Fall Detection [29] and Multicam Fall Dataset [10]	Accuracies Multicam (2 classes) 99.2% Multicam (2 classes) 99.25% UR FALL 99%
W. Min et al. [65]	2018	Skeleton joint tracking model provided by MS Kinect^®^ is used to estimate vertical/torso angle/depth characterization	SVM	Depth	TST Fall Detection [37]	Accuracy 92.05%
W. Min et al. [66]	2018	Object recognition through CNN and features of human shape sorted out as well as their spatial relations with furniture in the image/local characterization	Automatic engine classifier based on similarities (minimum quadratic error) between real-time actions and activity class features	RGB	This system-specific video dataset—no public access at revision time and UR Fall Detection [29]	Precision 94.44% Recall 94.95% Accuracy 95.5%
X. ShanShan et al. [67]	2018	Foreground extraction through background subtraction (GMM)/global characterization	SVM-radial kernel	RGB	Center For Digital Home Dataset– MMU [68]	Sensitivity 96.87% Accuracy 86.79%
Amal El Kaid et al. [69]	2019	Feature maps obtained through convolutional layers of a CNN/local characterization	Softmax based on features vector from CNN	RGB	This system-specific video dataset—no public access at revision time	Reduces false positives of angel assistance system by 17% by discarding positives assigned to people in a wheelchair
Chao Ma et al. [70]	2019	Face masking to preserve privacy and feature maps obtained through CNN/local characterization	Autoencoder SVM	RGB + IR	UR Fall Detection [29] and Multicam Fall Dataset [10] and Fall Detection Dataset [30] and This system-specific video Dataset—no public access at revision time	Autoencoder Sensitivity 93.3% Specificity 92.8% SVM Sensitivity 90.8% Specificity 89.6%
D. Kumar et al. [71]	2019	Silhouette segmentation by edge detection through HOG/global characterization + silhouette center angular velocity determined by long short-term memory (LSTM) model/local characterization	feature-threshold-based. Silhouette center point angular velocity	RGB	MOT Dataset [72] and UR Fall Detection [29] and COCO Dataset [73]	Accuracy 98.1%
F. Harrou et al. [74]	2019	Foreground extraction through background subtraction (direct comparison)/global characterization	SVM Linear kernel Polynomial kernel Radial kernel	RGB	UR Fall Detection [29] & Fall Detection Dataset [30]	Accuracy: Linear kernel 93.93% Polynomial kernel 94.35% Radial kernel 96.66%
J. Brieva et al. [75]	2019	Feature maps obtained through CNN from OF/ local characterization	Softmax based on features vector from CNN	RGB	This system-specific video dataset—no public access at revision time	Precision 95.27% Recall 95.42% F_1_ 95.34%
M. Hua et al. [76]	2019	Human keypoints identified by OpenPose (convolutional pose machines and human body vector construction) and recurrent neural network (RNN)-LSTM ANN used for pose prediction/local characterization	Fully connected layer	RGB	LE2I [23]	Precision 90.8% Recall 98.3% F_1_ 0.944
M. M. Hasan et al. [77]	2019	Human keypoints identified by OpenPose (convolutional pose machines and human body vector construction) and RNN-LSTM ANN/local characterization	Softmax based on features vector from RNN-LSTM	RGB	UR Fall Detection [29] & Fall Detection Dataset [30] & Multicam Fall Dataset [10]	URFD Sensitivity 99% Specificity 96% FDD Sensitivity 99% Specificity 97% Multicam Sensitivity 98% Specificity 96%
P. K. Soni et al. [78]	2019	Foreground extraction through background subtraction (GMM)/global characterization	SVM	RGB	UR Fall Detection [29]	Specificity 97.1% Sensitivity 98.15%
Ricardo Espinosa et al. [79]	2019	OF extracted from 1-s windows/global characterization + Feature maps obtained through CNN/local characterization	Softmax based on features vector from CNN SVM Random forest (RF) MLP KNN	RGB	UPFALL [80]	Sensitivity Softmax 97.95% SVM 14.1% RF 14.3% MLP 11.03% KNN 14.35%
S. Kalita et al. [81]	2019	BBs established in hands, head and legs through extended core9 framework/local characterization	SVM	RGB	UR Fall Detection [29]	Sensitivity 93.33% Specificity 95% Accuracy 94.28%
Saturnino Maldonado-Bascón et al. [82]	2019	Person detection through CNN YoLOv3 and feature extraction of the generated BB /local characterization	SVM	RGB	IASLAB-RGBD fallen person dataset [35] and This system-specific video dataset—no public access at revision time	Average results Precision 88.75% Recall 77.7%
X. Cai et al. [83]	2019	OF/global characterization + Wide residual network/local characterization	Softmax classifier implemented in the last layer of the ANN	RGB	UR Fall Detection [29]	accuracy 92.6%
Xiangbo Kong et al. [84]	2019	Segmentation by model provided by MS Kinect^®^ + depth map and CNN used for feature maps creation/depth characterization	Softmax based on features vector from CNN implemented in its last layer	Depth	This system-specific video dataset—no public access at revision time	Depending on the camera height accuracy, results between 80.1% and 100% are obtained
Xiangbo Kong et al. [85]	2019	Foreground extraction through background subtraction (Depth information) and HOG is calculated as a classifying feature	SVM-linear kernel	Depth	This system-specific video Dataset—no public access at revision time	Sensitivity 97.6% Specificity 100%
A. CARLIER et al. [86]	2020	Dense OF/global characterization + feature maps obtained through CNN/ local characterization	Fully connected layer	RGB	UR Fall Detection [29] and Multicam Fall Dataset [10] and LE2I [23]	Sensitivity 86.2% False discovery rate 11.6%
B. Wang et al. [87]	2020	Human keypoints identified by OpenPose (convolutional pose machines and human body vector construction) and followed by DeepSORT (CNN able to track numerous objects simultaneously)/local characterization	Classifiers are used to sort out falling state and fallen state Gradient boosted tree (GDBT) Decision tree (DT) RF SVM KNN MLP	RGB	UR Fall Detection [29] & Fall Detection Dataset [30] & LE2I [23]	F1-score Falling state GDBT 95.69% DT 84.85% RF 95.92% SVM 96.1% KNN 93.78% MLP 97.41% Fallen state GDBT 95.27% DT 95.45% RF 96.8% SVM 95.22% KNN 94.22% MLP 94.46%
C. Menacho et al. [88]	2020	Dense OF/global characterization and feature maps obtained through CNN/ local characterization	Fully connected layer	RGB	UR Fall Detection [29]	Accuracy 88.55%
C. Zhong et al. [89]	2020	Binarization based on IR threshold + edge identification/global characterization + feature maps obtained through convolutional layers of an ANN/local characterization	Based on features maps from CNN: Radial basis function neural network (RBFNN) SVM Softmax DT	IR	This system-specific video dataset—no public access at revision time	Multi-occupancy scenarios F1 score: RBFNN 89.57 (+/−0.62) SVM 88.74% (+/−1.75) Softmax 87.37% (+/−1.4) DT 88.9% (+/−0.68)
G. Sun et al. [90]	2020	pose estimation through OpenPose (convolutional pose machines and human body vector construction) and single-shot multibox detector-MobileNet (SSD-MobileNet)/local characterization	Support vector data description (SVDD) SVM KNN	RGB	COCO Dataset [73] and a specific video dataset—no public access at revision time	Sensitivity SVM 92.5% KNN 93.8% SVDD 94.6%
J. Liu et al. [91]	2020	Local binary pattern histograms from three orthogonal planes (LBP-TOP) applied over optical Flow after robust principal component analysis (RPCA) techniques have been applied over incoming video signals.	Sparse representations classification (SRC)	RGB	UR Fall Detection [29] & Fall Detection Dataset [30]	Accuracy: FDD dataset 98% URF dataset 99.2%
J. Thummala et al. [92]	2020	Foreground extraction through background subtraction (GMM)/global characterization	Feature-threshold-based. Object height/width ratio, ratio change speed and MHI.	RGB	LE2I [23]	Accuracy 95.16%
Jin Zhang et al. [93]	2020	Human keypoints identified by CNN (convolutional pose machines and human body vector construction)/local characterization	Logistic regression classifier based on: Rotation energy sequence Generalized force sequence	RGB	This system-specific video dataset—no public access at revision time	Fall detection rate 98.7% False alarm rate 1.05%
K. N. Kottar et al. [94]	2020	Segmentation through vibe [19] and illumination change-resistant algorithm (ICA) [95] then main silhouette axis determination	Feature-threshold-based. Silhouette main axis angle with vertical axis	RGB	This system-specific video dataset—no public access at revision time and PIROPO [96]	Specific database accuracy ICA—87%–96.34% VIBE—78.05%–86.5% PIROPO—ICA Walk accuracy 95% Seat accuracy 98.65%
Qi Feng et al. [97]	2020	Feature maps obtained through convolutional layers of a CNN and LSTM/local characterization	Softmax based on features vector from ANN implemented in its last layer	RGB	Multicam Fall Dataset [10] and UR Fall Detection [29] and this system-specific video dataset—no public access at revision time	Multicam Dataset Sensitivity 91.6% Specificity 93.5% UR Dataset Precision 94.8% Recall 91.4% THIS SYSTEM Dataset Precision 89.8% Recall 83.5%
Qingzhen Xu et al. [98]	2020	Human keypoints identified by OpenPose (convolutional pose machines and human body vector construction) and CNN used for feature maps creation/local characterization	Softmax based on features vector from CNN implemented in its last layer	RGB	UR Fall Detection [29] and Multicam Fall Dataset [10] and NTU RGB+D Dataset [99]	Accuracy rate 91.7%
Swe N. Htun et al. [100]	2020	Foreground extraction through background subtraction (GMM)/global characterization	Hidden Markov model (HMM) based onObservable data: Silhouette surface Centroid height Bounding box aspect ratio	RGB	LE2I [23]	Precision 99.05% Recall 98.37% Accuracy 99.8%
T. Kalinga et al. [101]	2020	Skeleton joint tracking model provided by MS Kinect^®^ is used to determine joint speeds and angles of different body parts/depth characterization	Feature-threshold-based. Joint speeds and angles of body parts	Depth	This system-specific video dataset—no public access at revision time	Accuracy 92.5% Sensitivity 95.45% Specificity 88%
Weiming Chen et al. [102]	2020	Human keypoints identified by OpenPose (convolutional pose machines and human body vector construction)/local characterization	Feature-threshold-based Hip vertical velocity Spine/ground plane angle BB aspect ratio	RGB	This system-specific video dataset—no public access at revision time	Accuracy 97% Sensitivity 98.3% Specificity 95%
X. Cai et al. [103]	2020	Feature maps obtained through hourglass convolutional auto-encoder (HCAE) ANN/local characterization	Softmax based on features vector from HCAE	RGB	UR Fall Detection [29]	Sensitivity 100% Specificity 93% Accuracy 96.2%
Y. Chen et al. [104]	2020	Foreground extraction through CNN and Bi-LSTM ANN/local characterization	Softmax based on features vector from RNN-Bi-LSTM	RGB	UR Fall Detection [29] and This system-specific video dataset—no public access at revision time	URFD Precision 0.897 Recall 0.813 F_1_ 0.852 Specific dataset Precision 0.981 Recall 0.923 F_1_ 0.948
Yuxi Chen et al. [105]	2020	Feature maps obtained through 3 different CNNs (LeNet, AlexNet y GoogLeNet)/depth characterization	Classification made by fully connected last layers of CNNs	Depth	Video dataset developed for the system in [84]	Average values Lenet Sensitivity 82.78% Specificity 98.07% AlexNet Sensitivity 86.84% Specificity 98.41% GoogLeNet Sensitivity 92.87% Specificity 99%
X. Wang et al. [106]	2020	Feature maps obtained through convolutional layers of an ANN/local characterization	Logistic function to identify frame-by-frame two classes in the prediction layer (person and fallen)	RGB	UR Fall Detection [29] & Fall Detection Dataset [30]	Average precision (AP) for fallen 0.97 mean average precision (mAP) for both classes 0.83

**Table 2 sensors-21-00947-t002:** System performance comparison.

Reference	Year	Input Signal	ANN/Classifiers and Performance					
C. -J. Chong et al. [6]	2015	RGB	Method 1 BB aspect ratio and CG position					
			Sensitivity 66.7%					
			Specificity 80%					
			Method 2 Ellipse orientation and aspect ratio + MHI					
			Sensitivity 72.2%					
			Specificity 90%					
F. Harrou et al. [28]	2017	RGB		Accuracy	Sensitivity		Specificity	
			KNN	91.94%	100%		86.00%	
			ANN	95.15%	100%		91.00%	
			NB	93.55%	100%		88.60%	
			MEWMA-SVM	96.66%	100%		94.93%	
Y. M. Galvão et al. [48]	2017	RGB	F1 score					
			Multilayer perceptron (MLP) 0.991					
			K-nearest neighbors (KNN) 0.988					
			SVM—polynomial kernel 0.988					
Leila Panahi et al. [60]	2018	Depth	Average results					
			SVM					
			Sensitivity 98.52%					
			Specificity 97.35%					
			Threshold-based decision					
			Sensitivity 98.52%					
			Specificity 97.35%					
K. Sehairi et al. [58]	2018	RGB	Accuracy					
			SVM-RBF 99.27%					
			KNN 98.91%					
			ANN 99.61%					
Chao Ma et al. [70]	2019	RGB+IR	Autoencoder					
			Sensitivity 93.3%					
			Specificity 92.8%					
			SVM					
			Sensitivity 90.8%					
			Specificity 89.6%					
F. Harrou et al. [74]	2019	RGB	Accuracy:					
			K-NN 91.94%					
			ANN 95.16%					
			Naïve Bayes 93.55%					
			Decision tree 90.48%					
			SVM 96.66%					
Ricardo Espinosa et al. [79]	2019	RGB			Sensitivity		Specificity	
			Softmax		97.95%		83.08%	
			SVM		14.10%		90.03%	
			RF		14.30%		91.26%	
			MLP		11.03%		93.65%	
			KNN		14.35%		90.96%	
Xiangbo Kong et al. [84]	2019	Depth		HOG+SVM	LeNet	AlexNet	GoogLeNet	ETDA-Net
			Average accuracy	89.48%	88.28%	93.53%	96.59%	95.66%
			Average specificity	95.43%	97.18%	97.56%	98.76%	99.35%
			Average sensitivity	83.75%	74.54%	87.10%	88.74%	91.87%
B. Wang et al. [87]	2020	RGB	F1 score					
			Falling state					
			GDBT 95.69%					
			DT 84.85%					
			RF 95.92%					
			SVM 96.1%					
			KNN 93.78%					
			MLP 97.41%					
			Fallen state					
			GDBT 95.27%					
			DT 95.45%					
			RF 96.8%					
			SVM 95.22%					
			KNN 94.22%					
			MLP 94.46%					
C. Zhong et al. [89]	2020	IR	F1 score					
			RBFNN 89.57 (+/−0.62)					
			SVM 88.74% (+/−1.75)					
			Softmax 87.37% (+/−1.4)					
			DT 88.9% (+/−0.68)					
C. Menacho et al. [88]	2020	RGB	Accuracy					
			VGG-16 87.81%					
			VGG-19 88.66%					
			Inception V3 92.57%					
			ResNet50 92.57%					
			Xception 92.57%					
			ANN proposed in this system 88.55%					
G. Sun et al. [90]	2020	RGB			Sensitivity		Specificity	
			SVM		92.50%		93.70%	
			KNN		93.80%		92.30%	
			SVDD		94.60%		93.80%	
Yuxi Chen et al. [105]	2020	Depth	Average values					
			Lenet					
			Sensitivity 82.78%					
			Specificity 98.07%					
			AlexNet					
			Sensitivity 86.84%					
			Specificity 98.41%					
			GoogLeNet					
			Sensitivity 92.87%					
			Specificity 99%					

**Table 3 sensors-21-00947-t003:** System performance evaluation datasets.

Signal Type	Dataset Name	Characteristics
Accelerometric and electroencephalogram (EEG) and RGB and passive infrared (IR)	Upfall [80]	17 volunteers execute falls and activities of daily life (ADL) of different types recorded by an accelerometer, EEG, RGB and passive IR systems
Depth and Accelerometric	Depth and accelerometric dataset [43]	Volunteers execute several activities, and falls are recorded by a depth system and accelerometers.
	TST fall detection [37]	11 volunteers execute 4 fall types and 4 ADLs recorded by RGB-depth (RGB-D) and accelerometer systems
	UR fall detection [29]	30 falls and 40 ADLs recorded by RGB-D and accelerometer systems
RGB	Center for digital home data set—MMU [68]	20 videos, including 31 falls and several ADLs
	LE2I [23]	191 different activities, including ADLs and 143 falls
	Charfi2012 dataset [25]	250 video sequences in four different locations, 192 containing falls, and 57 containing ADLs. Actors, under different light conditions, move in environments where occlusion exits and cluttered and textured background is common
	High-quality dataset [53]	It is a fall detection dataset that attempts to approach the quality of a real-life fall dataset. It has realistic settings and fall scenarios. In detail, 55 fall scenarios and 17 normal activity scenarios were filmed by five web-cameras in a room similar to one in a nursing home
	Multicam fall dataset [10]	The video data set is composed of several simulated normal daily activities and falls viewed from 8 different cameras and performed by one subject in 24 scenarios
	Simple fall detection dataset [20]	The dataset contains 30 daily activities such as walking, sitting down, squatting down, and 21 fall activities such as forward falls, backward falls and sideway falls
	MO dataset [72]	MOT dataset intends to be a framework for the fair evaluation of multiple people tracking algorithms. In this framework, the designers provide: Detections for all the sequences; A common evaluation tool providing several measures, from recall to precision to running time; An easy way to compare the performance of state-of-the-art tracking methods; Several challenges with subsets of data for specific tasks such as 3D tracking and surveillance.
	COCO dataset [73]	COCO is a large-scale object detection, segmentation, and captioning dataset designed to show common objects in context
	Piropo [96]	Multiple activities recorded in two different scenarios with both conventional and fish eye cameras
Depth	IASLAB-RGB fallen person dataset [35]	It consists of several static and dynamic sequences with 15 different people and 2 different environments
	Multimodal multiview dataset of human activities [50]	It consists of 2 datasets recorded simultaneously by 2 Kinect systems including ADLs and falls in a living room equipped with a bed, a cupboard, a chair and surrounding office objects illuminated by neon lamps on the ceiling or by sunlight
	Sdufall [12]	10 volunteers develop 6 activities recorded by RGB-D systems
	Falling detection [38]	6 volunteers perform 26 falls and similar activities recorded by RGB-D systems.
	Fall detection dataset [30]	5 volunteers execute 5 different types of fall
	NTU RGB+ dataset [99]	It is a large-scale dataset for human action recognition. It contains 56,880 action samples and includes 4 different modalities of data for each sample: RGB videos, depth map sequences, 3D skeletal data and IR videos
Synthetic Movement Databases	CMU Graphics Lab—motion capture library [55]	Library that captures synthetic movements through movement capture (MoCap) technology

## Data Availability

Not applicable.

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
