# Peer review of "Comprehensive Review of Vision-Based Fall Detection Systems"

_sensors, 2021, doi:10.3390/s21030947_

Round 1
Reviewer 1 Report
This work titled “Comprehensive review on vision-based fall detection systems” presents a comprehensive revision of all published articles in the main scientific data bases regarding this area, taken into consideration most of the literature published from 2015 to 2020.
In the research, the authors firstly introduced the problems they intended to solve by fall detection systems, secondly, they explain the methodology used for the paper gathering.
The authors present in table 1 the paper selection for fall detection systems based on computer vision; in the column of performance there is an inconsistency for the data presentation, e.g. in the rows of “F. Harrou et al. [27]”; “G. Sun et al. [92]”, the measures related to a different classification process are presented in two nested columns, but in other columns, similar measures for classification processes are presented in a different scheme, the authors need to improve the data presentation.
In the table 1 the acronym CG is used for the first time, then used along the paper without the proper definition, there are other acronyms that need to be reviewed to explain their meaning to the readers.
“quality approach, to quantify their intensity in an ulterior quantity approach.” …qualitative approach, quantitative approach
In this work there is another inconsistency in the presentation, e.g. in some parts the definition for the acronyms start with capital letters, and other used lower case letters, e.g. Motion History Image (MHI), Motion Co-occurrence Feature (MCF), robust principal component analysis (RPCA), etc. the authors should make a review and use consistency in writing.
Soft development kit (SDK)?
The author should make a review for math characters, e.g. “In other terms, given Xi as a class”, “and previous n values are”, “for a specific feature all k closest”, etc. it may be better if authors used math characters.
In general terms the authors should make a major review in the tables, and a significant proof reading, some ideas can be presented in a more proper manner.
Reviewer 2 Report
The content is very comprehensive, and it would be better if more illustration can be added by figures.
Reviewer 3 Report
The method used in the study is innovative. The method is of great importance for the elderly. The system can greatly improve safety in the home environment and improve care health care of the elderly. The article is well arranged and maintains the desired journal structure. The purpose of the research is clearly defined. There should be a clarification of who could use the results of this study in practice and what are the detailed perspectives research. The topic presented in the work is important and very topical. This is why societies all over the world are aging the point discussed in this article is important. This topic can be useful for developers who will prepare prevention systems falls and create vision systems to detect falls. It is worth emphasizing that the topic discussed in the article recognizes the problem of community needs older people and includes recommendations for developers. The number of technologies used to detect falls is large, and scientists have developed a huge number of systems that can work with them cooperate. It is important to improve systems and create solutions that will be used in practice. The authors made an attempt verification of systems in terms of their suitability. The research is quite advanced. The authors made significant contributions to literature review on fall detection systems based on vision, published in 2015-2020. In my opinion, the topic has been described very extensively, it is worth asking yourself whether it would not be better to divide it into two parts. The manuscript describes the needs elderly community and recommendations for developers are given. The conclusion is quite short, it can be expanded as the topic is topical and interesting. It is worth emphasizing who can use such systems. In conclusion, I recommend that you accept this manuscript with minor changes, in particular by increasing depth and scope discussion.
Reviewer 4 Report
- The following interesting review about wearable sensors for fall detection 10.3390/s18051613 may be discussed in the introduction to describe the wearable sensor field.
- Please provide the structure of the article at the end of the introduction paragraph, specifying the content of the following sections.
- All three tables are difficult to read. It would be better to structure them differently, maybe using some histograms to highlight the most interesting results.
- Insert a block scheme of the adopted search strategy for the papers selection.
- Italicize all paragraph titles
Round 2
Reviewer 1 Report
No comments now